# Cdc48 regulates a deubiquitylase cascade critical for mitochondrial fusion

Tânia Simões[1†], Ramona Schuster[1†], Fabian den Brave[2], Mafalda Escobar-Henriques[1]*

[1]Institute for Genetics, Cologne Excellence Cluster on Cellular Stress Responses in Aging-Associated Diseases, University of Cologne, Cologne, Germany; [2]Department of Molecular Cell Biology, Max Planck Institute of Biochemistry, Am Klopferspitz 18, Martinsried, Germany

**Abstract** Cdc48/p97, a ubiquitin-selective chaperone, orchestrates the function of E3 ligases and deubiquitylases (DUBs). Here, we identify a new function of Cdc48 in ubiquitin-dependent regulation of mitochondrial dynamics. The DUBs Ubp12 and Ubp2 exert opposing effects on mitochondrial fusion and cleave different ubiquitin chains on the mitofusin Fzo1. We demonstrate that Cdc48 integrates the activities of these two DUBs, which are themselves ubiquitylated. First, Cdc48 promotes proteolysis of Ubp12, stabilizing pro-fusion ubiquitylation on Fzo1. Second, loss of Ubp12 stabilizes Ubp2 and thereby facilitates removal of ubiquitin chains on Fzo1 inhibiting fusion. Thus, Cdc48 synergistically regulates the ubiquitylation status of Fzo1, allowing to control the balance between activation or repression of mitochondrial fusion. In conclusion, we unravel a new cascade of ubiquitylation events, comprising Cdc48 and two DUBs, fine-tuning the fusogenic activity of Fzo1.

DOI: https://doi.org/10.7554/eLife.30015.001

*For correspondence:
mafalda.escobar@uni-koeln.de

[†]These authors contributed equally to this work

**Competing interests:** The authors declare that no competing interests exist.

## Introduction

Mitochondria are dynamic organelles constantly undergoing fusion and fission events, modulated by a variety of post-translational modifiers including ubiquitin (*Escobar-Henriques and Langer, 2014*; *Komander and Rape, 2012*). Due to their pathological relevance, *e.g.* for Parkinson's disease, these processes are subject to intense investigation. For example, Parkin-dependent ubiquitylation of mitochondrial outer membrane (OM) proteins modulates the elimination of the damaged organelles by mitophagy, or *via* mitochondrial-derived vesicles (MDV) that fuse with the *late* endosome (*Pickrell and Youle, 2015*; *Sugiura et al., 2014*). Most fusion processes, including the Parkin-MDV pathway, rely on SNAREs (*McLelland et al., 2016*). In contrast, fusion of the endoplasmic reticulum (ER) and of mitochondria depend on large dynamin-related GTPases (*Escobar-Henriques and Anton, 2013*; *Hu and Rapoport, 2016*). In mitochondria, they are named mitofusins (Mfn1/Mfn2 in mammals, Fzo1 in yeast). Deficiencies in Mfn2 cause the type 2 subset of the Charcot-Marie-Tooth disease (CMT), the most common degenerative disorder of the peripheral nervous system (*Züchner et al., 2004*).

The ubiquitin-specific chaperone Cdc48/p97 is required to maintain mitochondrial morphology (*Esaki and Ogura, 2012*). However, the underlying molecular mechanism of how Cdc48 regulates mitochondrial dynamics is not understood. Cdc48 is an essential AAA-ATPase and one of the most abundant proteins in the cell, which recognizes many ubiquitylated substrates and is involved in a myriad of biological processes (*Franz et al., 2014*; *Meyer and Weihl, 2014*). Cdc48 segregates ubiquitylated substrates from protein complexes, or from membranes, thus allowing their proteolysis by the proteasome (*Franz et al., 2014*). For example, Cdc48 is important for ER-associated protein degradation (ERAD), modulates the turnover of mitochondrial OM proteins (OMMAD), participates

**eLife digest** Mitochondria are little compartments within a cell that produce the energy needed for most biological processes. Each cell possesses several mitochondria, which can fuse together and then break again into smaller units. This fusion process is essential for cellular health.

Two proteins in the cell have a major role in controlling mitochondrial fusion: Ubp12 and Ubp2. Ubp12 prevents fusion, while Ubp2 activates it. These molecules carry out their roles by acting on a third protein called mitofusin, which is a key gatekeeper of the fusion mechanism.

Cells often 'tag' proteins with small molecules called ubiquitin to change the protein's role and how it interacts with other cellular structures. Depending on how they are 'tagged', mitofusins can exist in two forms. One type of tagging means that the protein then promotes fusion of the mitochondria; the other leads to the mitofusin being destroyed by the cell.

It is still unclear how Ubp12, Ubp2 and the different forms of mitofusins interact with each other to finely control mitochondrial fusion. Here, Simões, Schuster et al. clarify these interactions in yeast and show how these proteins are themselves regulated.

Ubp2 promotes fusion by attaching to the mitofusin that is labeled to be destroyed, and removing this tag: the mitofusin will then not be degraded, and can promote fusion. Ubp12 prevents fusion through two mechanisms. First, it can remove the 'pro-fusion' tag on the mitofusin that prompts mitochondrial fusion. Second, Simões, Schuster et al. now show that Ubp12 also inhibits Ubp2 and its fusion-promoting activity.

In turn, the experiments reveal that a master protein called Cdc48 can control the entire Ubp12-Ubp2-mitofusin pathway. Cdc48 directly represses Ubp12 and therefore its anti-fusion activity. This inhibition also leaves Ubp2 free to stimulate fusion through its action on mitofusin.

The molecules involved in controlling mitochondrial fusion in yeast are very similar to the ones in people. In humans, improper regulation of mitofusins causes an incurable disease of the nerves and the brain called Charcot-Marie-Tooth 2A. Understanding how the fusion of mitochondria is controlled can lead to new drug discoveries.

DOI: https://doi.org/10.7554/eLife.30015.002

in apoptosis responses (*Laun et al., 2001*) and mediates clearance of damaged lysosomes by autophagy (*Avci and Lemberg, 2015*; *Heo et al., 2010*; *Papadopoulos et al., 2017*; *Tanaka et al., 2010*; *Wu et al., 2016*; *Xu et al., 2011*; *Zattas and Hochstrasser, 2015*). On the other hand, Cdc48 also binds E3 ubiquitin ligases and deubiquitylases (DUBs) thereby regulating substrate ubiquitylation (*Meyer and Weihl, 2014*).

DUBs are proteases that catalyze the reversion of the ubiquitylation reaction (*Love et al., 2007*), critically contributing to ubiquitin homeostasis (*Amerik and Hochstrasser, 2004*; *Kimura and Tanaka, 2010*; *Park and Ryu, 2014*; *Swatek and Komander, 2016*). DUBs activate ubiquitin by releasing it from ubiquitin precursor polypeptides but are also determinants for the modification status of ubiquitylated substrates, allowing to dampen ubiquitin-mediated events (*Clague et al., 2013*). Importantly, DUBs are associated with a number of human diseases and represent promising drug targets, whose regulation and mechanism of action need to be explored (*Heideker and Wertz, 2015*; *Sahtoe and Sixma, 2015*). Two deubiquitylases, Ubp2 and Ubp12, were found to have opposite effects on mitochondrial morphology (*Anton et al., 2013*). Ubiquitin chains on Fzo1 that are recognized and cleaved by Ubp12 activate mitochondrial fusion. In contrast, other ubiquitin chains on Fzo1 that instead are recognized and cleaved by Ubp2 target Fzo1 for proteasomal degradation and inhibit mitochondrial fusion. Therefore, although it is clear that ubiquitin is a double-faced regulator of mitochondrial fusion (*Escobar-Henriques and Langer, 2014*), how Ubp2 and Ubp12 exert opposite effects on Fzo1 and mitochondrial fusion remained poorly studied.

Here, we identify a role of Cdc48 in mitochondrial fusion, as part of a novel enzymatic cascade consisting of Cdc48, Ubp12 and Ubp2. Cdc48 negatively regulates Ubp12, which negatively regulates Ubp2, explaining why these two DUBs exert opposite effects on their targets and on ubiquitin homeostasis.

## Results

### Cdc48 promotes mitochondrial fusion and prevents Fzo1 turnover

Although it is clear that Cdc48 affects mitochondrial dynamics (*Esaki and Ogura, 2012*), the underlying mechanisms are unclear. The role of Cdc48 for mitochondrial morphology was investigated in the hypomorphic mutant *cdc48-2*, expressing GFP targeted to mitochondria. In this allele, Cdc48 is mutated for A547T, in its ATPase domain D2, whereas in the most commonly used *cdc48-3* strain, Cdc48 is instead mutated in R387K, in the D1 ATPase (C. Hickey and M. Hochstrasser, p. communication). Both *cdc48-3* and *cdc48-2* mutations impair typical Cdc48-dependent processes for transmembrane proteins, like ERAD (*Bays et al., 2001*; *Hitchcock et al., 2001*; *Latterich et al., 1995*). We observed that *cdc48-2* cells presented fragmented mitochondria (*Figure 1A*), consistent with the mitochondrial phenotypes observed upon impairment of the ATPase activity of Cdc48 (*Esaki and Ogura, 2012*). This suggested problems in mitochondrial fusion and prompted us to evaluate the role of Cdc48 on Fzo1, present at the outer membrane of mitochondria. Mitochondrial fusion is abolished in the absence of Fzo1 ubiquitylation (*Anton et al., 2013*). Consistent with mitochondrial fragmentation, we observed a decrease of Fzo1 ubiquitylation in *cdc48-2* mutant cells, when compared to wild-type (wt) cells (*Figure 1B*, black arrows). We have previously shown that pro-fusion ubiquitylation of Fzo1 increases its stability (*Anton et al., 2013*). Accordingly, the steady state levels of Fzo1 and its ubiquitylated forms were decreased in *cdc48-2* cells (compare *Figure 1C and B*), to a similar and not significantly different extent (data not shown). Consistent with the *cdc48-2* allele, the levels of Fzo1 were slightly decreased in the *cdc48-3* mutant or in cells deleted for the Cdc48 co-factors Npl4, Ufd1 and Ufd3/Doa1 (*Figure 1—figure supplement 1A–C*). It was previously shown that Ubc6, an endoplasmic reticulum (ER) membrane protein, is degraded by the proteasome *via* ERAD, a process dependent on Cdc48 (*Lenk et al., 2002*). Therefore, we also analyzed the steady state levels of Ubc6 in the same *CDC48* mutant strains. As expected, and in contrast to Fzo1, the steady state levels of Ubc6 were increased upon impairment of Cdc48 activity (*Figure 1C* and *Figure 1—figure supplement 1A–C*). This suggested that Cdc48 regulates Fzo1 by a mechanism different from OMMAD or ERAD. Since both Fzo1 and Ubc6 were mostly affected in the *cdc48-2* mutant, we decided to use this strain for further analysis. However, it is unclear why *cdc48-2* affects Ubc6 and Fzo1 stronger than *cdc48-3*. We investigated why *cdc48-2* mutant cells have lower levels of Fzo1, by testing with cycloheximide (CHX) chase experiments if Cdc48 regulates Fzo1 stability. Moreover, to simultaneously test the role of the proteasome, we deleted the efflux pumps Snq2 and Pdr5. We observed that Fzo1 degradation was inhibited by the presence of the proteasome inhibitor MG132, indicating that the decreased levels of Fzo1 observed in *cdc48-2* cells were due to proteasome-dependent turnover of Fzo1 (*Figure 1D*). In contrast, proteasome inhibition did not affect Fzo1 turnover in wt cells consistent with previous observations (*Anton et al., 2013*; *Escobar-Henriques et al., 2006*). Importantly, all these phenotypes could be rescued by expression of the wt Cdc48 protein but not by expression of the Cdc48$^{A547T}$ variant, mimicking the specific mutation in *cdc48-2* (*Figure 1—figure supplement 2A–C*). In conclusion, Cdc48 is required to maintain the Fzo1 protein, thus promoting mitochondrial fusion events.

### Cdc48 binds and regulates ubiquitylated Fzo1

We further investigated how Cdc48 affected Fzo1. Given that stress conditions disrupt mitochondrial tubulation (*Knorre et al., 2013*), it was important to show that Cdc48 directly regulates Fzo1 and mitochondrial morphology. First, co-immunoprecipitation experiments revealed that Cdc48 physically interacted with Fzo1 (*Figure 2A*). We previously showed that the formation of ubiquitin chains on Fzo1 (*Figure 2A*, black arrows), which are linked to lysine 398, requires previous ubiquitylation of its lysine 464 (*Anton et al., 2013*). Therefore, Fzo1 ubiquitylation is lost in the mutant Fzo1$^{K464R}$ (*Figure 2A*). We observed that the interaction between Cdc48 and the non-ubiquitylated variant Fzo1$^{K464R}$ was impaired (*Figure 2A*), in agreement with ubiquitin being recognized by Cdc48. To assess the specificity of the *cdc48-2* effect on Fzo1 protein levels, we tested if this depended on Fzo1 ubiquitylation. Thus, the non-ubiquitylated variant Fzo1$^{K464R}$ was used. We observed that the steady state levels of Fzo1$^{K464R}$ were largely insensitive to the *cdc48-2* mutation (*Figure 2—figure supplement 1*). This points to a direct regulatory role of Cdc48 on Fzo1, only after its ubiquitylation. These pro-fusion ubiquitin forms on Fzo1 are recognized by Ubp12. In addition, we previously

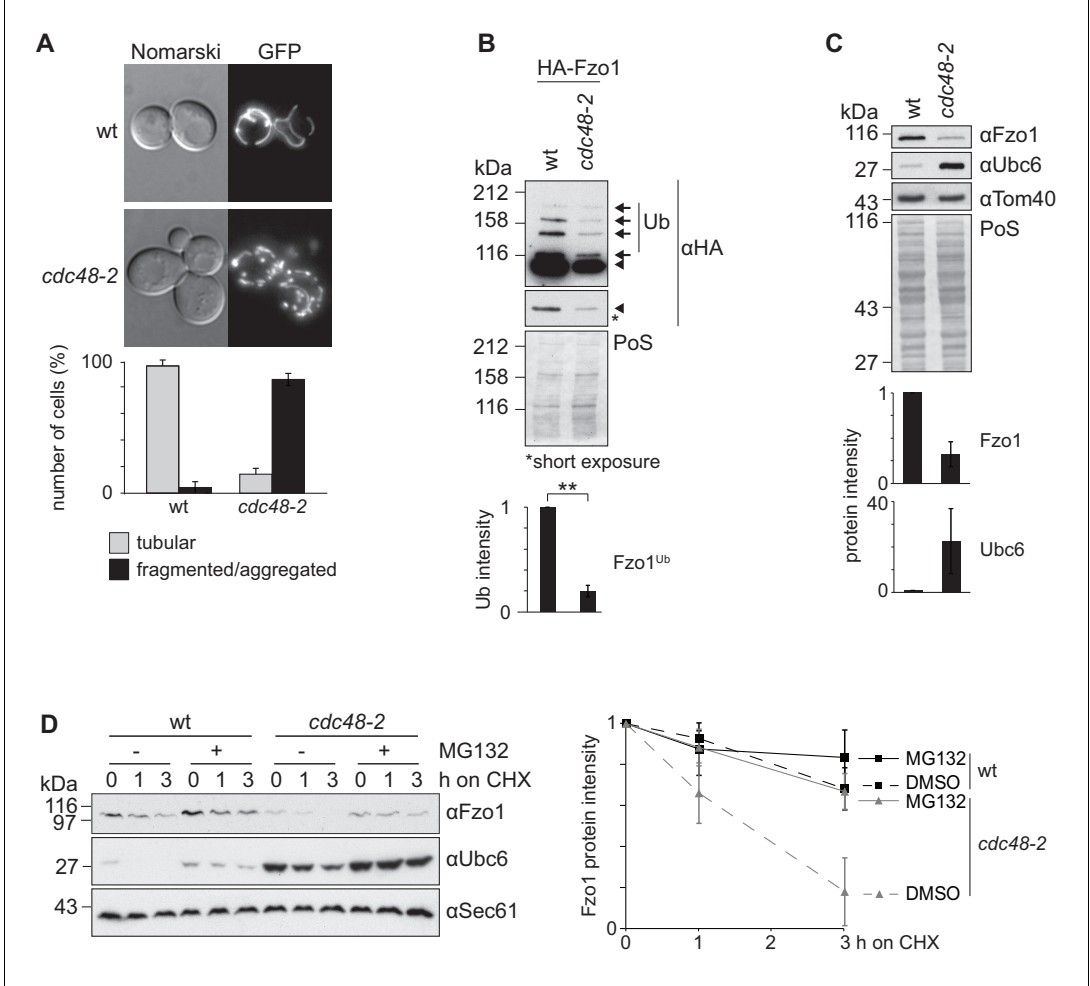

**Figure 1.** Cdc48 regulates Fzo1 and mitochondrial fusion. (**A**) Mitochondrial morphology of *CDC48* mutant cells. Wild-type (wt) or *cdc48-2* mutant cells were analyzed for mitochondrial tubulation after expressing a mitochondrial-targeted GFP plasmid. Cellular (Nomarski) and mitochondrial (GFP) morphology were visualized by fluorescence microscopy. Bottom panel, quantification of four independent experiments (with more than 200 cells each) including mean and standard deviation (SD), as described (*Cumming et al., 2007*). (**B**) Ubiquitylation of Fzo1 upon mutation of *CDC48*. Crude mitochondrial extracts from wt or *cdc48-2* mutant cells expressing HA-Fzo1, or the corresponding empty vector, were solubilized and analyzed by SDS-PAGE and immunoblotting using HA-specific antibodies. Unmodified and ubiquitylated forms of HA-Fzo1 are indicated by a black arrowhead or black arrows, respectively. Ubiquitylated forms of Fzo1 are labeled with Ub. Bottom panel, quantification of three independent experiments, normalized to PoS and including SD. **, p≤0.01 (paired t-test). (**C**) Steady state levels of Fzo1 upon mutation of *CDC48*. Total cellular extracts of wt or *cdc48-2* mutant cells were analyzed by SDS-PAGE and immunoblotting using Fzo1- or Ubc6-specific and, as a loading control, Tom40-specific antibodies. Bottom panels, quantification of three independent experiments, including SD. (**D**) Proteasome dependence of Fzo1 degradation in *cdc48-2* mutant cells. The turnover of endogenous Fzo1 expressed in *Δpdr5 Δsnq2* and *Δpdr5 Δsnq2 cdc48-2* cells was assessed after inhibition of cytosolic protein synthesis with cycloheximide (CHX), for the indicated time points in exponentially growing cultures in absence or presence of the proteasomal inhibitor MG132. Samples were analyzed by SDS-PAGE and immunoblotting using Fzo1-specific, Ubc6-specific (as an unstable protein control) and Sec61-specific (as a loading control) antibodies. Right panel, quantification of five independent experiments, including SD. PoS, PonceauS staining.

DOI: https://doi.org/10.7554/eLife.30015.003

The following figure supplements are available for figure 1:

**Figure supplement 1.** Cdc48 regulates Fzo1 and mitochondrial fusion.
DOI: https://doi.org/10.7554/eLife.30015.004

**Figure supplement 2.** Cdc48 regulates Fzo1 and mitochondrial fusion.
DOI: https://doi.org/10.7554/eLife.30015.005

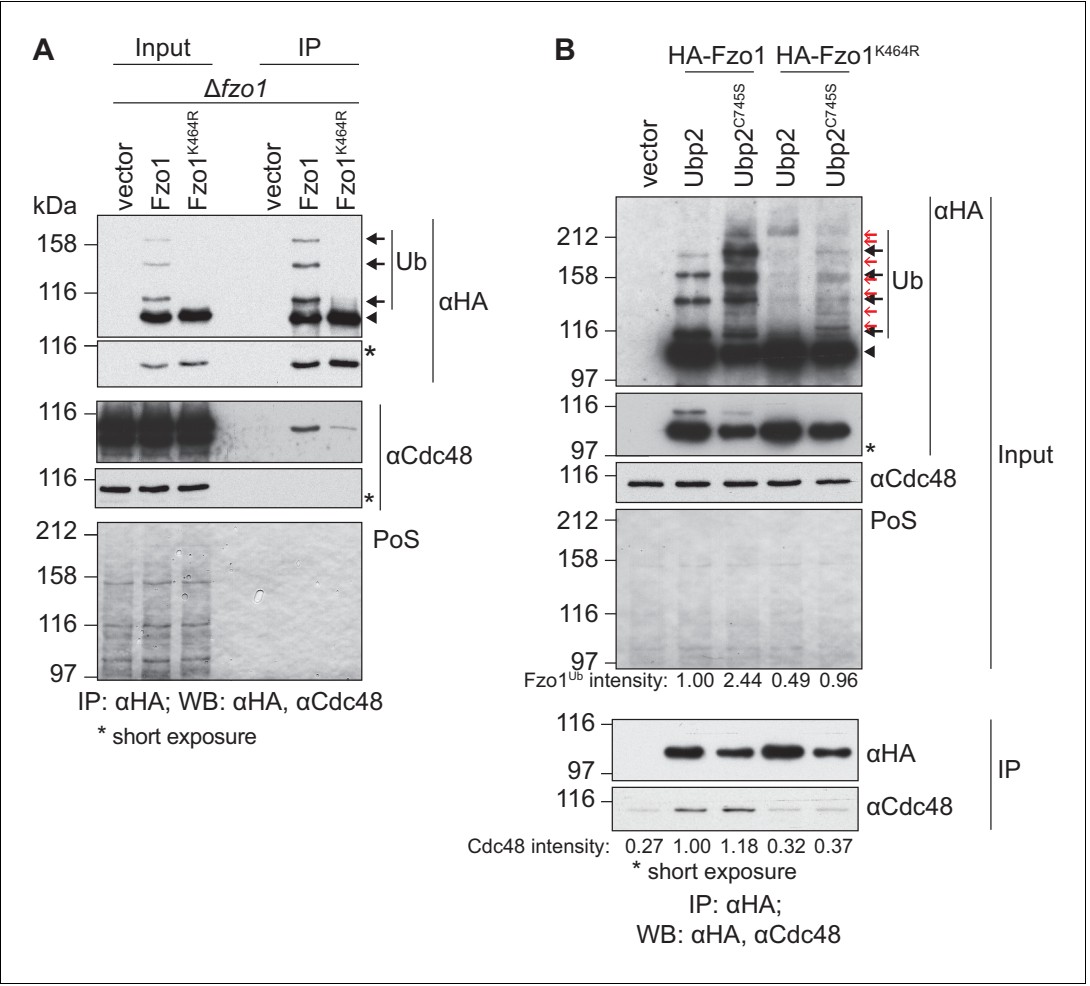

**Figure 2.** Cdc48 specifically affects ubiquitylated Fzo1. (**A**) Physical interaction between Cdc48 and ubiquitylated Fzo1. HA-Fzo1, HA-Fzo1$^{K464R}$ or the corresponding vector were expressed in $\Delta fzo1$ cells. Crude mitochondrial extracts were lysed and HA-tagged Fzo1 was precipitated using HA-coupled beads and analyzed by SDS-PAGE and immunoblotting using HA- and Cdc48-specific antibodies. Unmodified and ubiquitylated forms of HA-Fzo1 are indicated as in 1B. (**B**) Effect of the anti-fusion ubiquitylation of Fzo1 on its interaction with Cdc48. HA-Fzo1 or HA-Fzo1$^{K464R}$, expressed in the presence of Ubp2 ($\Delta fzo1$ cells plus empty vector) or Ubp2$^{C745S}$ ($\Delta ubp2$ $\Delta fzo1$ cells plus Ubp2$^{C745S}$-Flag), or the corresponding vector control (the empty vectors corresponding to HA-Fzo1 and Ubp2$^{C745S}$-Flag, expressed in $\Delta ubp2$ $\Delta fzo1$ cells), were analyzed for Cdc48 interaction, as in 2A. Unmodified and ubiquitylated forms of HA-Fzo1 are indicated by a black arrowhead or black arrows, respectively. Red arrows with no fill indicate Fzo1 ubiquitylated species specifically accumulating upon expression of Ubp2$^{C745S}$. PoS, PonceauS staining; IP, immunoprecipitation; WB, western blot.

DOI: https://doi.org/10.7554/eLife.30015.006

The following figure supplement is available for figure 2:

**Figure supplement 1.** Cdc48 specifically affects ubiquitylated Fzo1.
DOI: https://doi.org/10.7554/eLife.30015.007

identified other ubiquitin forms on Fzo1, that inhibit fusion. They are removed by Ubp2 and can be detected only in the presence of the catalytically inactive variant Ubp2$^{C745S}$ (**Anton et al., 2013**) (**Figure 2B**, Input, red arrows). Therefore, we investigated binding of Cdc48 to Fzo1 under these conditions, where both pro-fusion and anti-fusion forms are present. We noticed that despite the clear increase in ubiquitylation of Fzo1 upon Ubp2$^{C745S}$ expression (2.44 times), Cdc48 binding to Fzo1 was not increased (**Figure 2B**). Therefore, the additional presence of ubiquitin chains inhibiting fusion does not increase Cdc48 binding. Consistently, for the Fzo1$^{K464R}$ variant, which in the presence of Ubp2$^{C745S}$ is ubiquitylated to a similar level as the wt protein (0.96 times, despite the

absence of pro-fusion ubiquitylation), no binding to Cdc48 above background can be detected. Thus, similar to Ubp12, Cdc48 recognizes specifically the pro-fusion ubiquitylated forms of Fzo1.

## Cdc48 supports turnover of ubiquitylated Ubp12

Given the specific interaction of both Cdc48 (*Figure 2B*) and Ubp12 (*Anton et al., 2013*) with ubiquitin chains on Fzo1 promoting fusion, we tested if Cdc48 regulated Ubp12. To analyze if Ubp12 is an unstable protein, wt and *cdc48-2* cells were transformed with an episomal plasmid expressing Ubp12 under the *ADH1* promoter (*Anton et al., 2013*). CHX chase experiments revealed that Ubp12 is degraded in a Cdc48- and proteasome-dependent manner (*Figure 3—figure supplement 1A and B*). Similarly, chromosomally tagged Ubp12 is an unstable protein and its turnover depends on Cdc48 (*Figure 3A*). To analyze if Ubp12 is ubiquitylated, the DUB was immunoprecipitated and analyzed by immunoblotting for Ubp12-Flag or for ubiquitin (*Figure 3B*). We observed slower migrating forms of Ubp12 with the Flag-specific antibody, which were also detected by a ubiquitin-specific antibody. These studies demonstrated that Ubp12 is modified by ubiquitin. We next tested whether Cdc48 could be co-immunoprecipitated with Ubp12, from solubilized crude mitochondrial extracts. We observed that Ubp12 physically interacted with Cdc48 (*Figure 3C*), suggesting that Cdc48 directly supports degradation of ubiquitylated Ubp12.

## Cdc48 regulation of Fzo1 depends on Ubp12

Our results show that Cdc48 and Ubp12 have opposing roles on Fzo1 ubiquitylation levels (*Figure 1B* and [*Anton et al., 2013*]). Consistently, Ubp12 and Cdc48 also present opposing phenotypes regarding mitochondrial tubulation (*Figure 1A* and [*Anton et al., 2013*]). Given that Cdc48 controls Ubp12 levels, we speculated that Cdc48 regulates mitochondrial morphology and Fzo1 *via* Ubp12. We monitored mitochondrial morphology in *cdc48-2* cells in presence or absence of *UBP12*, expressing mitochondrial-targeted GFP. Strikingly, deletion of *UBP12* in *cdc48-2* cells rescued mitochondrial tubulation, resembling Δ*ubp12* cells (*Figure 4A*). Importantly, the mitochondrial hypertubulation of Δ*ubp12* cells depended on Fzo1 (*Figure 4—figure supplement 1A–C*). Even in Δ*fzo1* Δ*dnm1* cells, resembling wt cells in mitochondrial shape, further deletion of *UBP12* did not induce hypertubulation, confirming that Ubp12 regulates mitochondrial morphology *via* Fzo1 (*Figure 4— figure supplement 1D*). Mitochondrial fusion is also required to maintain the cellular growth on respiratory media, *i.e.* media containing the non-fermentable carbon sources glycerol or lactate (*Hermann et al., 1998*). Therefore, to further support the physiological importance of Cdc48 and Ubp12, we analyzed the respiratory capacity of *cdc48-2* in presence or absence of *UBP12*. In agreement with restored tubulation of mitochondria, we observed that the growth defect of *cdc48-2* cells at 37°C on lactate media could be improved upon deletion of *UBP12* (*Figure 4B*). Given that Δ*fzo1* cells irreversibly loose mitochondrial DNA, we investigated if this is also the case for *cdc48-2* cells. Consistent with the respiratory reversibility of *cdc48-2* cells upon further deletion of *UBP12*, we observed that *cdc48-2* cells did not lose mitochondrial DNA (*Figure 4—figure supplement 2A and B*). Importantly, the respiratory defect of *cdc48-2* cells could be complemented by expression of Cdc48 but not of Cdc48$^{A547T}$ (*Figure 4—figure supplement 2C*). Finally, *cdc48-2*Δ*ubp12* cells also showed improved ubiquitylation of Fzo1 (*Figure 4C*). Together, these results show that Cdc48 maintains Fzo1 ubiquitylation and activates mitochondrial fusion by downregulating Ubp12. However, two pieces of evidence suggest that Cdc48 might have other functions in this pathway, apart from regulating Ubp12. First, we observed that the physical interaction between Fzo1 and Cdc48 is not mediated by Ubp12 (*Figure 4—figure supplement 2D*), suggesting that Cdc48 directly recognizes ubiquitylated Fzo1. Second, deletion of *UBP12* in *cdc48-2* cells did not restore the steady state levels of Fzo1 (*Figure 4—figure supplement 2E*). Notably, this is consistent with our previous observation that mitochondrial fusion depends on ubiquitylated rather than on the steady state levels of Fzo1 (*Anton et al., 2013*).

## Ubp12 mediates deubiquitylation of Ubp2

We noticed that increased levels of Fzo1, present in Δ*ubp12* cells, specifically depended on Ubp2 (*Figure 5A*). Therefore, Ubp12 and Ubp2, which affect the stability of Fzo1 in opposite manners, are also interdependent. Next, we analyzed if Ubp2 and Ubp12 also presented other opposing and interdependent phenotypes related to ubiquitin. First, we analyzed cellular growth of cells lacking

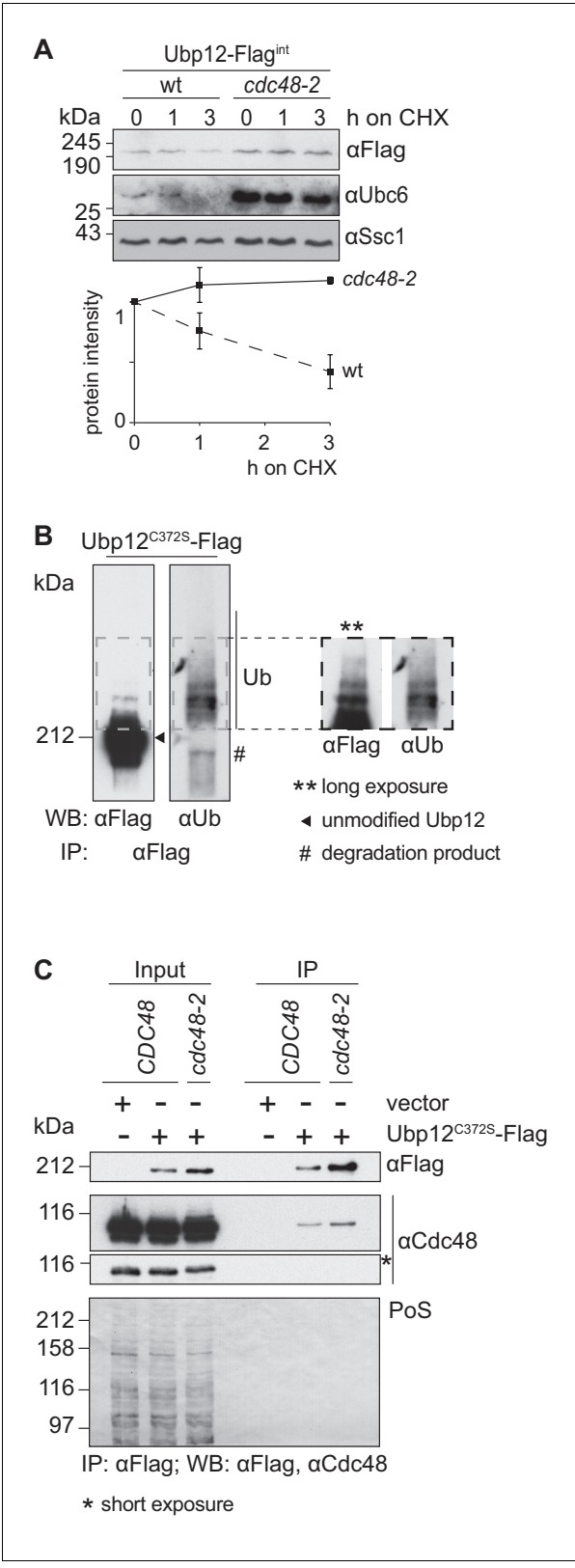

**Figure 3.** Cdc48 supports ubiquitin-dependent turnover of Ubp12. (**A**) Stability of the Ubp12 protein. The turnover of Ubp12 endogenously Flag tagged (Ubp12-Flag$^{int}$), in wt or *cdc48-2* cells, was assessed with CHX chase, as in 1D. Samples were analyzed by SDS-PAGE and immunoblotting using a Flag-, Tom40- and, as an unstable protein control, a Ubc6-specific antibody. Bottom panel, quantification of three independent experiments, including SD. *Figure 3 continued on next page*

*Figure 3 continued*

(B) Ubiquitylation of Ubp12. The Ubp12$^{C372S}$-Flag inactive variant, expressed from an episomal plasmid, was immunoprecipitated from total soluble extracts using Flag-coupled beads. After elution, Ubp12 was analyzed by western blot using Flag- or ubiquitin (Ub - P4D1)-specific antibodies. Ubiquitylated forms of Ubp12$^{C372S}$-Flag are labeled with Ub. (C) Physical interaction between Cdc48 and Ubp12. The catalytically inactive Ubp12$^{C372S}$-Flag variant, expressed from an episomal plasmid, or the corresponding empty vector, were expressed in Δ*ubp12* (*CDC48*) or Δ*ubp12 cdc48-2* (*cdc48-2*) mutant cells and analyzed for Cdc48 interaction. Crude mitochondrial extracts were lysed, Flag-tagged Ubp12 was precipitated using Flag-coupled beads, and the eluate analyzed by SDS-PAGE and immunoblotting using Flag- and Cdc48-specific antibodies. PoS, Ponceau S staining; IP, immunoprecipitation; WB, western blot.

DOI: https://doi.org/10.7554/eLife.30015.008

The following figure supplement is available for figure 3:

**Figure supplement 1.** Cdc48 supports ubiquitin-dependent turnover of Ubp12.

DOI: https://doi.org/10.7554/eLife.30015.009

---

*UBP2, UBP12* or both, in the presence of sub-lethal doses of CHX, a phenotype commonly tested to monitor imbalances in ubiquitin homeostasis (*Gerlinger et al., 1997*; *Hanna et al., 2003*; *Rumpf and Jentsch, 2006*). Second, we directly quantified the levels of free ubiquitin *vs.* substrate-conjugated ubiquitin in the same strains. We observed that indeed Ubp2 and Ubp12 had opposite phenotypes (*Figure 5—figure supplement 1*). In addition, the consistent interdependence of these two enzymes suggested a DUB hierarchy, which prompted us to test a possible regulation of the Ubp2 protein by Ubp12. We tested if Ubp2 is an unstable protein and whether Ubp12 is involved in its degradation, after inhibition of protein synthesis with CHX. The levels of genomically tagged Ubp2 decreased over time and Ubp2-turnover was regulated by Ubp12 (*Figure 5B*) and by the proteasome (*Figure 5—figure supplement 2A*). Moreover, co-immunoprecipitation experiments revealed that Ubp2 interacted with Ubp12, suggesting a direct regulation between both DUBs (*Figure 5—figure supplement 2B*). We therefore investigated if Ubp2 could be ubiquitylated, in a Ubp12-dependent manner. After immunoprecipitation of Ubp2-Flag, and consistent with recent observations (*Cavellini et al., 2017*), we observed the presence of slowly migrating forms of Ubp2 during electrophoresis, in wt cells (*Figure 5—figure supplement 2C*) but mostly in Δ*ubp12* cells (*Figure 5C*, left panel). Importantly, we show that these forms could also be detected using a ubiquitin-specific antibody, demonstrating that they represent ubiquitylated Ubp2 (*Figure 5C* and *Figure 5—figure supplement 2C*, right panels). This indicates that Ubp12 mediates deubiquitylation of Ubp2 and suggests that Ubp2 acts downstream of Ubp12, thus revealing a hierarchical cascade between DUBs, of relevance for the protein levels of Fzo1 and for ubiquitin homeostasis.

## Ubp12 recognizes short K48-linked ubiquitin chains on Fzo1

In contrast to numerous proteins that are destabilized in absence of DUBs, deletion of *UBP12* stabilizes Fzo1 (*Figure 6—figure supplement 1*) and Ubp2 (*Figure 5B*). Consistently, the two other known substrates of Ubp12 – Rad23 (*Gödderz et al., 2017*) and Gpa1 (*Wang et al., 2005*) are also not destabilized in Δ*ubp12* cells. To characterize the deubiquitylation reaction of Ubp12 in more detail, we analyzed the ubiquitin linkages on Fzo1 and Ubp2 accumulating in Δ*ubp12* cells. Overexpression of ubiquitin mutated in K48R strongly decreased Fzo1 and Ubp2 ubiquitylation, revealing that their ubiquitin chains are linked *via* K48 (*Figure 6A and C*). However, the ubiquitin chains on Fzo1 that destabilize it and inhibit fusion, which are not bound by Ubp12, are also K48-linked (*Figure 6B*) (*Anton et al., 2013*). Thus, differences in ubiquitin chains cannot explain why Ubp12 stabilizes its substrates. To further analyze Ubp12, its ubiquitin chain preference was tested using *in vitro* deubiquitylation assays (*Hospenthal et al., 2015*). As a substrate, we used either K48-linked or K63-linked ubiquitin, present in the form of either di-ubiquitin (*Figure 6D*) or ubiquitin chains (*Figure 6E*). However, in all cases, Ubp12 revealed no chain preference (*Figure 6D,E*). This suggested that it is not Ubp12 but rather the chains themselves on the substrates that prevent their turnover. Thus, we determined the number of ubiquitin moieties present on Fzo1, upon co-expression of tagged and non-tagged ubiquitin molecules. We observed that co-expression of ubiquitin and Myc-ubiquitin decomposed the first ubiquitylated form of Fzo1, *i.e* running closest to non-modified Fzo1, into two bands (*Figure 6F*). This corresponds to the presence of either ubiquitin or Myc-

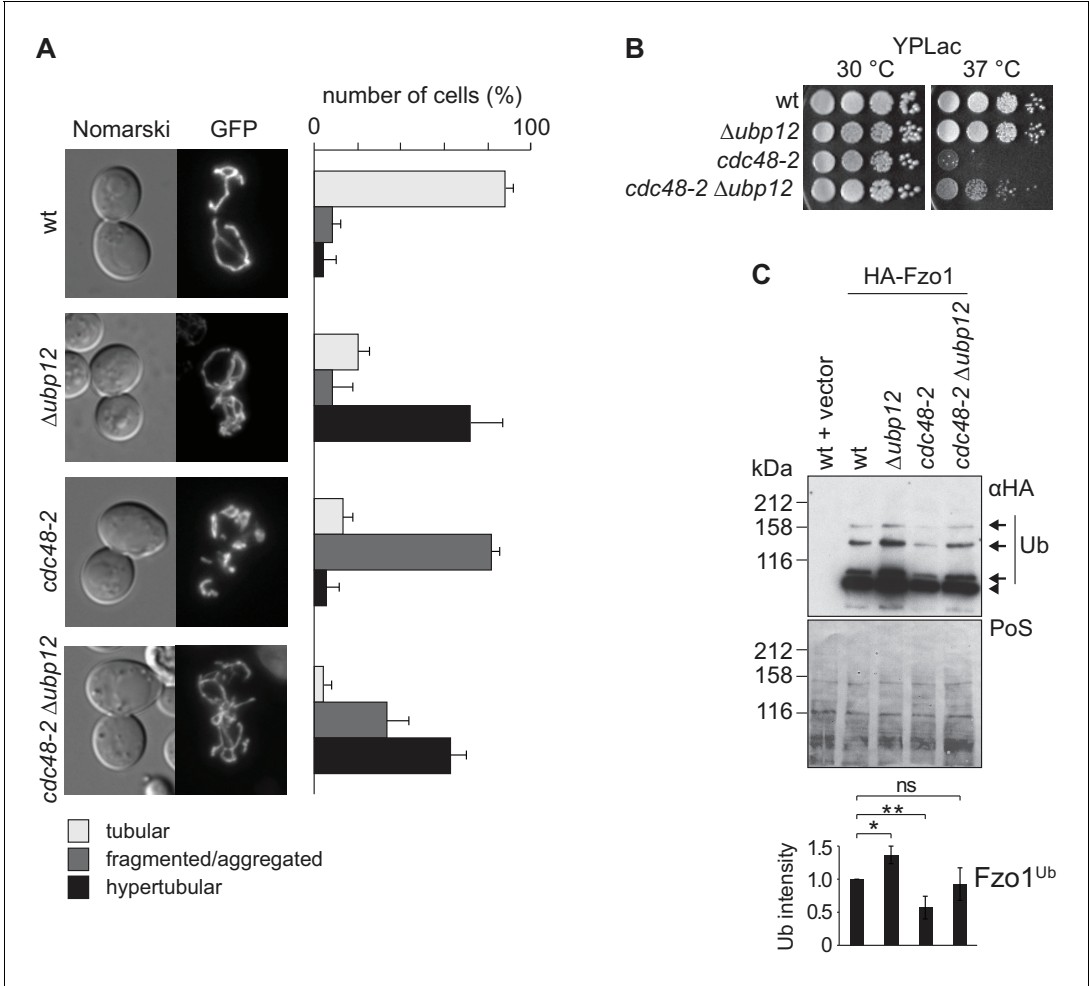

**Figure 4.** Interdependence of Cdc48 and Ubp12 for Fzo1 regulation. (**A**) Mitochondrial morphology upon deletion of *UBP12* and/or mutation of *CDC48*. The indicated mutant cells were analyzed for mitochondrial tubulation after expressing a mitochondrial-targeted GFP plasmid, as in *Figure 1A*. Right panel, quantification from three different experiments (with more than 200 cells each), including SD, as described (*Cumming et al., 2007*) (**B**) Respiratory capacity of cells upon deletion of *UBP12* and/or mutation of *CDC48*. Fivefold serial dilutions of exponentially growing cells of wt or the mutant strains Δ*ubp12*, *cdc48-2*, and Δ*ubp12 cdc48-2* were spotted on YP media supplemented with lactate (YPLac) and incubated at 30°C for two days or 37°C for five days. (**C**) Ubiquitylation levels of Fzo1 upon deletion of *UBP12* and/or mutation of *CDC48*. Crude mitochondrial extracts from the indicated strains additionally expressing HA-Fzo1, or the corresponding empty vector, were analyzed by SDS-PAGE and immunoblotting using an HA-specific antibody. Unmodified and ubiquitylated forms of HA-Fzo1 are indicated as in *Figure 1B*. Bottom panel, quantification of four independent experiments, normalized to PoS and including SD. ns, $p > 0.05$. *, $p \leq 0.05$, **, $p \leq 0.01$ (One-way ANOVA, Tukey's multiple comparison test). PoS, PonceauS staining.

DOI: https://doi.org/10.7554/eLife.30015.010

The following figure supplements are available for figure 4:

**Figure supplement 1.** Interdependence of Cdc48 and Ubp12 for Fzo1 regulation.
DOI: https://doi.org/10.7554/eLife.30015.011

**Figure supplement 2.** Interdependence of Cdc48 und Ubp12 for Fzo1 regulation.
DOI: https://doi.org/10.7554/eLife.30015.012

ubiquitin attached to Fzo1 and confirms that this form corresponds to mono-ubiquitylated Fzo1. Interestingly, however, for the two other ubiquitylated forms with lower electrophoretic mobility, we observed that only two additional bands could be observed above each of them. They correspond to either the presence of two Myc-ubiquitin molecules or one ubiquitin and one Myc-ubiquitin conjugated to Fzo1. These results suggest that the K48 chains on Fzo1 consist of two ubiquitin moieties. In conclusion, Ubp12 recognizes ubiquitylated chains on Fzo1 composed of a very small number of

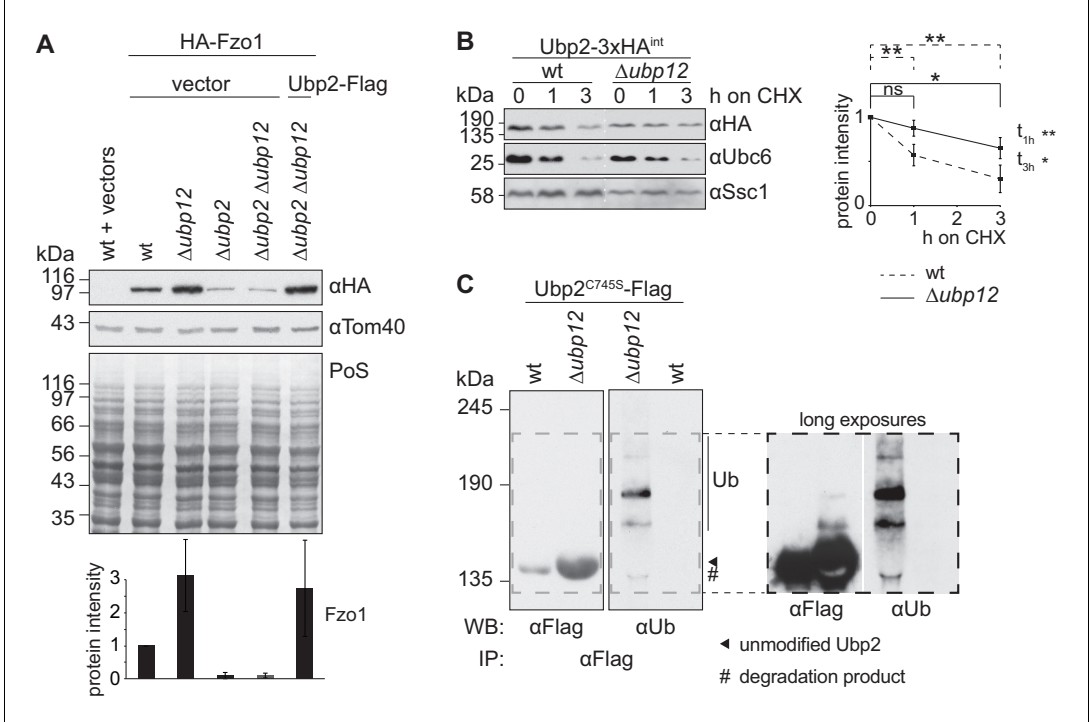

**Figure 5.** Ubp12 modulates Ubp2 ubiquitylation and turnover. (**A**) Interdependent role of Ubp2 and Ubp12 for the steady state levels of Fzo1. Total cellular extracts of wt or Δ*ubp2*, Δ*ubp12*, and Δ*ubp2* Δ*ubp12* mutant cells expressing HA-Fzo1 and also expressing either Ubp2-Flag or the corresponding empty vector, as indicated, were analyzed by SDS-PAGE and immunoblotting using HA- and Tom40-specific antibodies. Bottom panel, quantification of four independent experiments, including SD. (**B**) Turnover of endogenous Ubp2 in wt or Δ*ubp12* cells. The turnover of endogenously 3xHA-tagged Ubp2 (Ubp2-3xHA$^{int}$) was assessed as in 3A. Samples were analyzed by SDS-PAGE and immunoblotting using antibodies against HA, Ubc6 and Ssc1. Right panel, quantification of four independent experiments, including SD. For the statistical analysis of the degradation kinetics of each strain, a paired t-test was used; for the statistical analysis of the difference in steady state levels of both strains at the indicated time points ($t_{1h}$, $t_{3h}$) an unpaired t-test was used. ns, $p > 0.05$; *, $p \leq 0.05$; **, $p \leq 0.01$. (**C**) Ubiquitylation of Ubp2. The Ubp2$^{C745S}$-Flag inactive variant, expressed in wt or Δ*ubp12* cells, was immunoprecipitated from total soluble extracts using Flag-coupled beads. Eluted Ubp2 was analyzed by western blot using Flag- or ubiquitin (Ub - P4D1)-specific antibodies. Ubiquitylated forms of Ubp2$^{C745S}$-Flag are labeled with Ub. PoS, Ponceau S staining; IP, immunoprecipitation; WB, western blot.

DOI: https://doi.org/10.7554/eLife.30015.013

The following figure supplements are available for figure 5:

**Figure supplement 1.** Ubp12 modulates Ubp2 ubiquitylation and turnover.
DOI: https://doi.org/10.7554/eLife.30015.014

**Figure supplement 2.** Ubp12 modulates Ubp2 ubiquitylation and turnover.
DOI: https://doi.org/10.7554/eLife.30015.015

ubiquitin moieties. We therefore propose that Ubp12 does not stabilize its substrates because their ubiquitin chains are too short to target proteasomal turnover.

## Ubp12-Ubp2 cascade activity impinges on Fzo1 ubiquitylation

Both Ubp12 and Ubp2 deubiquitylate Fzo1, but they clearly bind different forms of ubiquitylated Fzo1 (*Anton et al., 2013*). Ubp12 binds ubiquitylated forms of Fzo1 that stabilize Fzo1 and promote mitochondrial fusion. In turn, Ubp2 recognizes other ubiquitylated forms of Fzo1, that instead signal Fzo1 turnover thus preventing mitochondrial fusion. Given that Ubp12 acts upstream of Ubp2, we speculated that the pro-fusion ubiquitylated forms of Fzo1, Ubp12-specific, would also precede its Ubp2-specific anti-fusion forms. This predicts an impairment of anti-fusion forms in the absence of pro-fusion forms. Therefore, as previously, the mutant Fzo1$^{K464R}$ was chosen as a tool, because it loses the pro-fusion ubiquitylation (*Figure 7A*, inset, black arrows, compare lanes 1 and 2). Moreover, as in *Figure 2B*, the catalytically-inactive Ubp2$^{C745S}$ protein was expressed additionally. This allows visualization of the Ubp2-specific anti-fusion forms as well (*Figure 7A*, inset, red arrows, lane

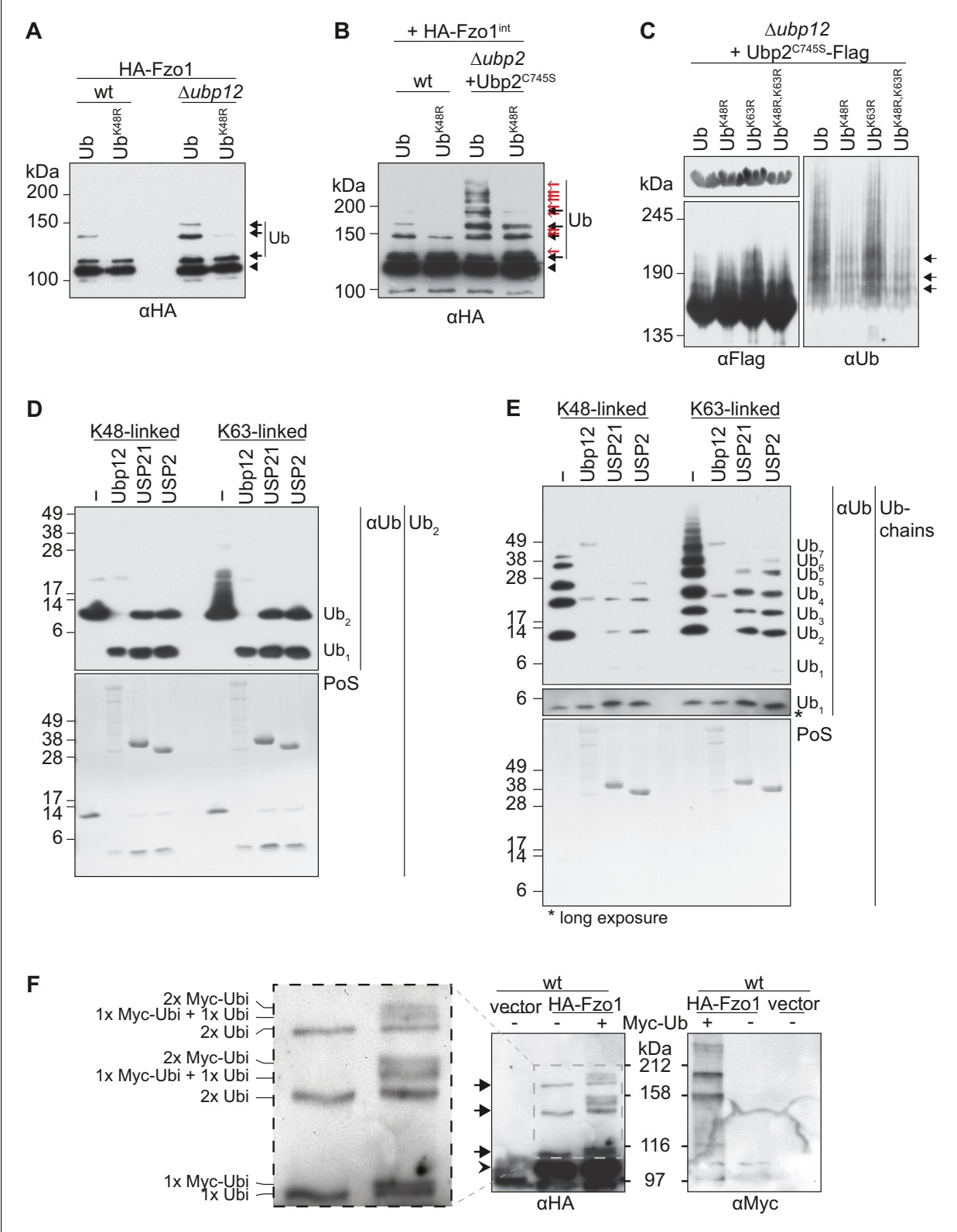

**Figure 6.** Characterization of the deubiquitylation reaction by Ubp12. (**A**) Analysis of ubiquitin chain-type composition of Fzo1. Crude mitochondrial extracts from wt or Δ*ubp12* mutant cells expressing HA-Fzo1, and over-expressing either wt ubiquitin (Ub) or ubiquitin with a K48R mutation (Ub^K48R), were solubilized, subjected to HA-immunoprecipitation and analyzed by SDS-PAGE and immunoblotting using an HA-specific antibody. Unmodified and ubiquitylated forms of HA-Fzo1 are indicated as in 1B. (**B**) Ubiquitin chain-type analysis of Fzo1 upon Ubp2^C745S expression. Crude mitochondrial

*Figure 6 continued on next page*

*Figure 6 continued*
extracts from wt or Δ*ubp2* (expressing Ubp2$^{C745S}$) cells expressing HA-Fzo1 endogenously, and overexpressing either wt ubiquitin (Ub) or Ub$^{K48R}$, were analyzed as in A. Unmodified and ubiquitylated forms of HA-Fzo1 are indicated as in 2B (C) Analysis of Ubp2 ubiquitin chain composition in Δ*ubp12* cells. Soluble extracts from Δ*ubp12* cells expressing Ubp2$^{C745S}$-Flag and different ubiquitin variants (as indicated) were prepared and Flag-tagged Ubp2$^{C745S}$ was precipitated using Flag-coupled beads. The eluate was analyzed by SDS-PAGE and immunoblotting using antibodies against Flag and ubiquitin (Ub; αP4D1). (D) Deubiquitylation (DUB) assay using Ub$_2$ chains. Purified di-ubiquitin chains (Ub$_2$) composed of either only K48- or K63-linkages were treated with the purified DUBs Ubp12, USP21 and USP2. Treated chains were analyzed by SDS-PAGE and immunoblotting using a ubiquitin-specific antibody (Ub; αP4D1). Mono-ubiquitin or di-ubiquitin chains are labeled with Ub$_1$ or Ub$_2$, respectively. (E) DUB assay using Ub-chains. Purified poly-ubiquitin chains (Ub-chains) composed of either only K48- or K63-linkages were treated with the purified DUBs Ubp12, USP21 or USP2. Treated chains were analyzed by SDS-PAGE and immunoblotting as in C. Ubiquitin chains were labeled as in D with the subscript value indicating the amount of ubiquitin moieties in the respective chain. (F) Ubiquitylation pattern of Fzo1. Wt cells expressing HA-Fzo1 were analyzed for Fzo1 ubiquitylation upon the expression of Myc-ubiquitin, or the respective empty vector. HA-Fzo1 was immunoprecipitated from mitochondrial extracts using HA-coupled beads. Eluted Fzo1 was split into two and samples were analyzed by SDS-PAGE and immunoblotting using HA- or Myc-specific antibodies. Unmodified and ubiquitylated forms of HA-Fzo1 are indicated as in 1B. The composition of the additional species apparent upon co-expression of Myc-tagged ubiquitin is explained in the inset. PoS, PonceauS staining.
DOI: https://doi.org/10.7554/eLife.30015.016
The following figure supplement is available for figure 6:

**Figure supplement 1.** Characterization of the deubiquitylation reaction by Ubp12.
DOI: https://doi.org/10.7554/eLife.30015.017

3), resulting in a massive increase in overall ubiquitylation of Fzo1 (compare lanes 1 and 3). As predicted by our hypothesis, much of this increase was lost when K464 was mutated to R (compare lanes 3 and 4). This shows that Ubp2-dependent ubiquitylation largely requires previous K464-dependent ubiquitylation . Therefore, pro-fusion ubiquitylation, which stabilizes Fzo1, primes Fzo1 for the formation of anti-fusion ubiquitylation. These anti-fusion forms, instead, signal Fzo1 for proteasomal degradation, so that in Δ*ubp2* cells Fzo1 is less abundant (*Anton et al., 2013*). Taking this into consideration, the steady state levels of Fzo1 were used as a read-out for the presence of anti-fusion ubiquitylation on Fzo1. We noticed that whereas the steady state levels of Fzo1 decreased by 91% inΔ*ubp2* cells, as expected, the steady state levels of Fzo1$^{K464R}$ only decreased by 47% (*Figure 7B*). This shows that Fzo1$^{K464R}$ is much less sensitive to the deletion of *UBP2* than wt Fzo1, consistent with a lower abundance of the anti-fusion ubiquitylation. To confirm this result, the levels of Fzo1 were also tested upon further deletion of *MDM30* inΔ*ubp2* cells, which encodes the E3 ligase-component responsible for pro-fusion ubiquitylation on Fzo1 (*Cohen et al., 2008*; *Escobar-Henriques et al., 2006*; *Fritz et al., 2003*). Indeed, we could observe a rescue of Fzo1 steady state levels inΔ*ubp2* Δ*mdm30* cells, confirming that pro-fusion precedes anti-fusion ubiquitylation on Fzo1 (*Figure 7C*). We conclude that Ubp2-specific ubiquitylation of Fzo1 largely depends on Ubp12-specific ubiquitylation of Fzo1, indicating a regulatory cascade of Ubp12 and Ubp2 on Fzo1.

## Cdc48 mitochondrial phenotypes depend on Ubp2

To challenge the Cdc48-DUBs regulatory cascade, we first tested if the role of Cdc48 on Fzo1 steady state levels depended on Ubp2 and Ubp12. Indeed, and in contrast to wt cells, in Δ*ubp2* Δ*ubp12* cells the steady state levels of Fzo1 were insensitive to further mutating Cdc48 (*Figure 8A*). Moreover, Δ*ubp2* cells and Δ*ubp2* Δ*ubp12* were similarly insensitive to the presence of the *cdc48-2* allele (*Figure 8B*), consistent with the *UBP2 UBP12* epistasis results (*Figure 5A* and *Figure 5—figure supplement 1A and B*). Next, we tested if overexpression of Ubp2 could rescue *cdc48-2* phenotypes. This was to be expected because deletion of *UBP12* rescues *CDC48* mutant phenotypes but also leads to increased levels of Ubp2. Consistently, mitochondrial tubulation was significantly improved under these conditions (*Figure 8C*). Moreover, Ubp2 overexpression improved the growth defect of *cdc48-2* cells on lactate media at the non-permissive temperature of 37°C, supporting the physiological impact of the Ubp2 levels in *cdc48-2* cells (*Figure 8D*). Therefore, the respiratory capacity of the *cdc48-2* cells could be improved not only by *UBP12* deletion but also by overexpression of Ubp2. Finally, a physical interaction between Ubp2 and Cdc48 could be observed (*Figure 8—figure supplement 1*). Together our results highlight a model in which Cdc48, Ubp12 and Ubp2 orchestrate a multilayered cascade regulation, culminating on Fzo1 ubiquitylation and mitochondrial fusion.

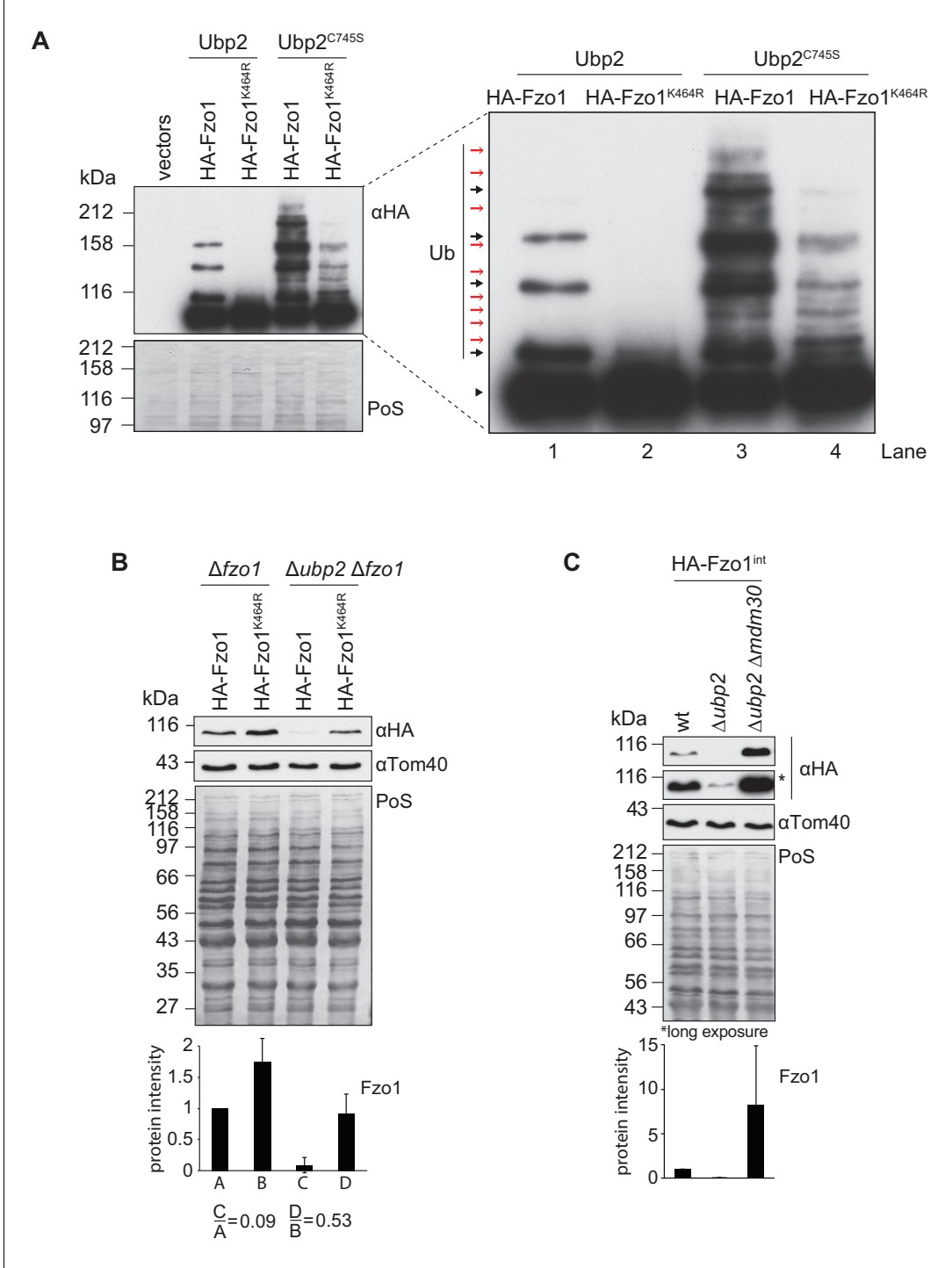

**Figure 7.** Interdependent roles of Ubp2 and Ubp12. (**A**) Effect of Ubp2$^{C745S}$ on Fzo1$^{K464R}$ ubiquitylation. HA-Fzo1 or HA-Fzo1$^{K464R}$ were expressed in the presence of Ubp2 (Δ*fzo1* cells plus empty vector) or instead in the presence of Ubp2$^{C745S}$ (Δ*ubp2* Δ*fzo1* plus Ubp2$^{C745S}$-Flag), as indicated. Crude mitochondrial extracts were solubilized and HA-tagged Fzo1 was analyzed by SDS-PAGE and immunoblotting using an HA-specific antibody. Unmodified and ubiquitylated forms of HA-Fzo1 are indicated as in 2B. (**B**) Effect of *UBP2* deletion on the steady state levels of Fzo1$^{K464R}$. Total cellular extracts of indicated strains expressing HA-Fzo1 or HA-Fzo1$^{K464R}$ as indicated were analyzed by SDS-PAGE and immunoblotting using HA- and Tom40-specific antibodies. Bottom panel, quantification of five independent experiments, including SD. (**C**) Effect of Ubp2 and Mdm30 on the steady state levels of Fzo1. Total cellular extracts of wt, Δ*ubp2* and Δ*ubp2* Δ*mdm30* cells expressing HA-tagged Fzo1 endogenously (HA-Fzo1$^{int}$) were analyzed by SDS-PAGE and immunoblotting using HA- and Tom40-specific antibodies. Bottom panel, quantification of three independent experiments, including SD. PoS, Ponceau S staining.

DOI: https://doi.org/10.7554/eLife.30015.018

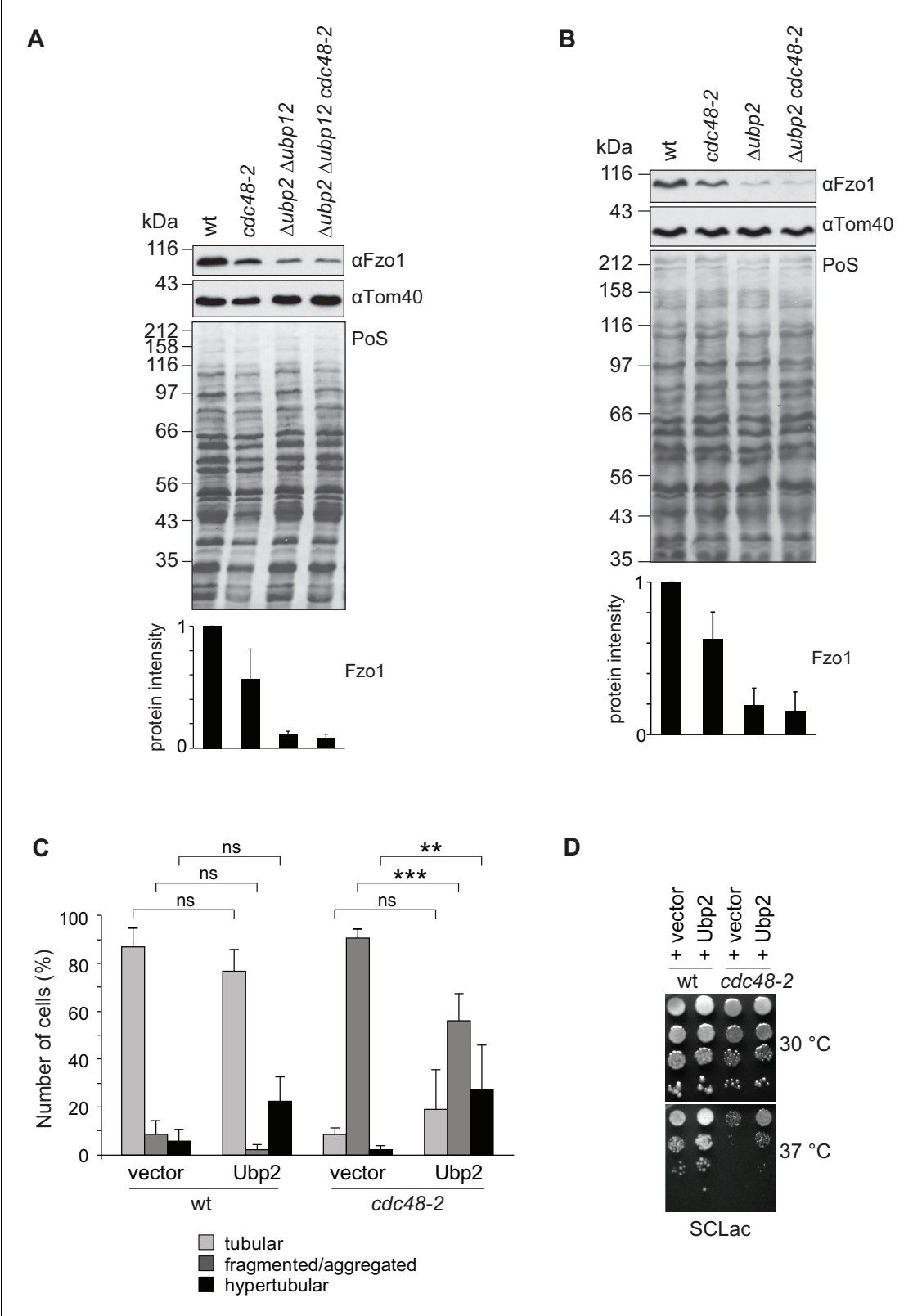

**Figure 8.** Cdc48 regulates mitochondrial fusion *via* Ubp12 and Ubp2. (**A**) Steady state levels of Fzo1 in Δ*ubp2* Δ*ubp12* upon mutation of *CDC48*. Total cellular extracts of wt, *cdc48-2*, Δ*ubp2* Δ*ubp12* and Δ*ubp2* Δ*ubp12 cdc48-2* cells were analyzed by SDS-PAGE and immunoblotting using Fzo1- and Tom40-specific antibodies. Bottom panel, quantification of five independent experiments, including SD. (**B**) Steady state levels of Fzo1 in Δ*ubp2* cells upon deletion of *CDC48*. Total cellular extracts of wt, *cdc48-2*, Δ*ubp2* and Δ*ubp2 cdc48-2* cells were analyzed by SDS-PAGE and immunoblotting using

*Figure 8 continued on next page*

*Figure 8 continued*
Fzo1- and Tom40-specific antibodies. Bottom panel, quantification of five independent experiments, including SD. (**C**) Mitochondrial morphology of *cdc48-2* cells upon overexpression of Ubp2. Wt or *cdc48-2* mutant cells expressing Ubp2 or the corresponding empty vector were analyzed for mitochondrial tubulation after expressing a mitochondrial-targeted GFP plasmid, as in *Figure 1A*. Quantification from three different experiments (with more than 200 cells each), including SE, as described (*Cumming et al., 2007*). ns, p>0.05. **p≤0.01, ***p≤0.001 (One-way ANOVA, Tukey's multiple comparison test). (**D**) Role of Ubp2 overexpression on the respiratory capacity of *CDC48*-deficient cells. A spot assay was performed as described in *Figure 4B* with the indicated cells but using synthetic media supplemented with lactate (SCLac) and incubated for 4 days. PoS, Ponceau S staining.
DOI: https://doi.org/10.7554/eLife.30015.019

The following figure supplement is available for figure 8:

**Figure supplement 1.** Cdc48 regulates mitochondrial fusion *via* Ubp12 and Ubp2.
DOI: https://doi.org/10.7554/eLife.30015.020

## Discussion

Precise regulation of cellular processes by protein ubiquitylation requires a tight control of the enzymes involved. We reveal a new mode of DUB regulation by Cdc48 for Fzo1 and mitochondrial fusion (*Figure 9*). This is likely of broader relevance for the regulation of DUBs and ubiquitin homeostasis.

### Synergistic function of Cdc48 in Fzo1 ubiquitylation

Cdc48 promotes degradation of Ubp12, controlling Fzo1 ubiquitylation. Ubp12 prevents mitochondrial fusion by two means. On the one hand, it removes the ubiquitylation on Fzo1 that is required for fusion. On the other hand, it promotes degradation of Ubp2. This leaves the anti-fusion ubiquitylation of Fzo1 unopposed, resulting in Fzo1 degradation. Therefore, by supporting turnover of Ubp12, Cdc48 dually preserves mitochondrial fusion events. In contrast, when only a non-functional variant of the protein is present, as is the case in *cdc48-2* cells, Cdc48 cannot protect the pro-fusion

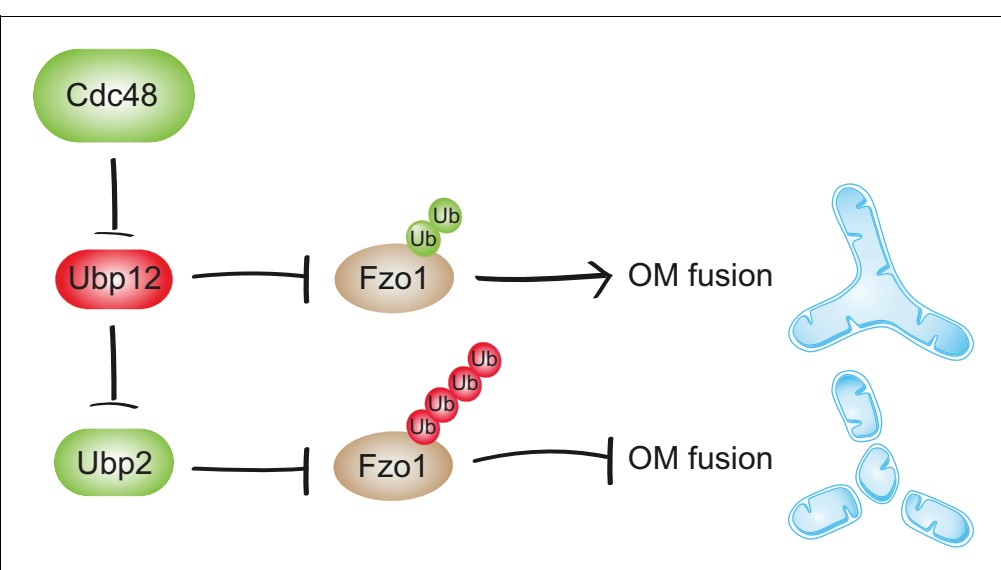

**Figure 9.** Synergistic regulation of mitochondrial fusion by the Cdc48 cascade. Cdc48 supports turnover of Ubp12, stabilizing ubiquitylation on Fzo1 that promotes mitochondrial fusion (green ubiquitins). Moreover, degradation of Ubp12 stabilizes Ubp2, facilitating the removal of ubiquitin chains on Fzo1 inhibiting mitochondrial fusion (red ubiquitins). Thereby, Cdc48 activates mitochondrial fusion *via* Ubp12 and Ubp2. In contrast, Cdc48 impairment blocks progression of mitochondrial fusion by actively preventing Ubp12 turnover. Ubp12 then leads to a cascade of events inhibiting mitochondrial fusion: A) removal of the pro-fusion ubiquitylated forms and B) inhibition of Ubp2, consequently leading to the accumulation of the anti-fusion ubiquitylated forms. This cascade allows a synergistic effect of Cdc48, *via* a DUB regulatory cascade, to effectively promote or inhibit mitochondrial fusion.
DOI: https://doi.org/10.7554/eLife.30015.021

ubiquitylation of Fzo1. In this case, the cascade will synergistically converge in degradation of Fzo1 and thus inhibition of mitochondrial fusion will occur. The interdependence between these two pathways contributes to a coordinated cellular decision by Cdc48 to either fuse mitochondria or instead prevent it by degrading Fzo1. Moreover, the Cdc48-Ubp12-Ubp2 cascade allows fine-tuning of substrate ubiquitylation and modulation of the biological processes thereof, as exemplified for Fzo1 and mitochondrial fusion (*Figure 9*).

## Roles of Cdc48 on mitochondrial dynamics

Cdc48/p97 extracts ubiquitylated substrates from membranes, thus allowing their recognition and degradation by the proteasome (*Franz et al., 2014*; *Rape et al., 2001*). This is exemplified with the ER protein Ubc6, and was also shown for mitochondrial OM proteins (*Neutzner et al., 2007*), including mitofusins under damaging conditions (*Tanaka et al., 2010*). Therefore, Cdc48/p97 and ubiquitin regulate mitochondrial fusion in both yeast and mammals. Moreover, eukaryotes present a similar ubiquitin pattern of mitofusins, suggesting that the new function of Cdc48 presented here could be conserved in mammals under non-damaging conditions.

## Critical role of the DUB cascade for mitochondrial fusion

Mitochondrial fusion is a complex multistep process dependent on sequential events involving GTP binding and hydrolysis by Fzo1, Fzo1 oligomerization and finally ubiquitylation of Fzo1 (*Anton et al., 2011*; *Brandt et al., 2016*; *Cohen et al., 2011*; *Ishihara et al., 2004*). Although it is clear that ubiquitin critically determines mitochondrial fusion events, the underlying mechanisms are largely unknown (*Anton et al., 2013*). The DUBs Ubp12 and Ubp2 cleave different ubiquitylated forms of Fzo1 that either promote or repress mitochondrial fusion, respectively (*Anton et al., 2013*). Here, given that Ubp12 regulates Ubp2, we show that these two ubiquitylation pathways are connected. Consistently, on Fzo1, Ubp12-specific ubiquitylation also precedes Ubp2-specific ubiquitylation. In fact, unopposed anti-fusion ubiquitylation, as it is the case in Δ*ubp2* cells, disrupts mitochondrial tubulation. This renders the role of Ubp2 in mitochondrial dynamics quite clear, namely protecting mitochondrial fusion. In contrast, the need for a dedicated DUB that removes the pro-fusion ubiquitylation forms, *i.e.* the need for Ubp12, remained unclear. Now, the Ubp12-Ubp2 cascade allows to understand the purpose of Ubp12, solving the paradox of why inhibition of the pro-fusion ubiquitylation on Fzo1 is required: in fact, too much pro-fusion ubiquitylation also means too much anti-fusion ubiquitylation, a problem counteracted by the deubiquitylation activity of Ubp12 on Fzo1. We conclude that this cascade ensures a tight control of Fzo1 ubiquitylation at levels sufficient to allow mitochondrial fusion but preventing unnecessary ubiquitylation that instead targets Fzo1 for proteasomal turnover.

## Which E3 ligases and DUBs modify Fzo1?

The cascade between Ubp12 and Ubp2 also allows revising recent results linking Ubp2 and Mdm30 (*Cavellini et al., 2017*). Mdm30 catalyzes the formation of the pro-fusion ubiquitin forms on Fzo1 (*Cohen et al., 2008*). The pro-fusion forms are bound and cleaved by Ubp12, depend on lysine 464 of Fzo1, and are essential for mitochondrial fusion (*Anton et al., 2013*). As to the anti-fusion ubiquitin forms on Fzo1, two types could now be observed: low molecular weight, K464-independent, anti-fusion ubiquitylation (as seen in *Figure 7A*, lane 4), consistent with previous results (*Anton et al., 2013*), but mostly high molecular weight anti-fusion ubiquitylation, instead K464-dependent (as seen in *Figure 7A*, lane 3). This shows that the anti-fusion ubiquitin forms on Fzo1 largely depend on its pro-fusion forms. Therefore, it is not surprising that anti-fusion, Ubp2-specific, ubiquitylation on Fzo1 also largely depends on Mdm30. Nevertheless, future studies are required to clarify if Mdm30 itself catalyzes the formation of this high molecular weight fraction of the anti-fusion ubiquitylation on Fzo1. Moreover, it is clear that Mdm30 is not the ligase responsible for the anti-fusion low molecular weight forms on Fzo1 (*Anton et al., 2013*), which therefore remains to be identified.

## Novel DUB cascade controlling ubiquitin homeostasis

Our results unravel for the first time a regulatory cascade of two DUBs, Ubp12 and Ubp2, with opposing functions in ubiquitin homeostasis. A 20–40% depletion in ubiquitin levels leads to cellular

growth defects under various stress conditions in yeast, to lethality or infertility in mice, and to neurological diseases like ataxia, gracile axonal dystrophy or Parkinson's disease (*Kimura and Tanaka, 2010*; *Park and Ryu, 2014*). The level of free ubiquitin is adjusted to the cellular needs, and is critically regulated by deubiquitylase activity (*Chernova et al., 2003*; *Swaminathan et al., 1999*). Here, we reveal distinct roles of two DUBs - Ubp2 and Ubp12 - for the maintenance of ubiquitin homeostasis. Δ*ubp12* cells are hyperresistant to cycloheximide (CHX), a chemical inhibitor of protein translation. Similar observations were previously reported in proteasome mutants, with impaired proteolysis (*Gerlinger et al., 1997*). Consistently, just like proteasome mutants, also Δ*ubp12* cells accumulate conjugated ubiquitin, without affecting the levels of free ubiquitin. In turn, Δ*ubp2* cells showed a 40% depletion of free ubiquitin and hypersensitivity to CHX, consistent with similar observations in strains presenting decreased free ubiquitin levels (*Hanna et al., 2003*). Nevertheless, along with reduced free ubiquitin, deletion of *UBP2* also clearly led to increased levels of ubiquitin conjugates, as observed upon DmUsp5 depletion in the fruit fly (*Kovács et al., 2015*). In fact, the importance of free ubiquitin pools versus ubiquitin conjugates for cellular growth is not well understood. Our analysis of Δ*ubp2* cells sheds light on this question, demonstrating that depletion of free ubiquitin is epistatic over the accumulation of ubiquitylated conjugates for cellular growth.

## Differences in DUB behavior

What could justify the opposite behavior of Ubp2 and Ubp12 in ubiquitin homeostasis and substrate turnover? The removal of ubiquitin from a substrate is generally expected to increase its stability, as observed for Fzo1 in Δ*ubp2* cells. Consistently, Ubp2 appears as a general quality control deubiquitylase recognizing both K48- and K63-linked ubiquitin chains that signal for turnover, both by the UPS and by the lysosome (*Anton et al., 2013*; *Fang et al., 2016*; *Ho et al., 2017*; *Silva et al., 2015*). In contrast, the turnover of both Fzo1 and Ubp2 is decreased in Δ*ubp12* cells. Moreover, Ubp12 does not stabilize Rad23 (*Gödderz et al., 2017*) and Gpa1 (*Wang et al., 2005*), *i.e.* its two other known substrates. Ubp12 exhibits a broad substrate specificity in vitro recognizing both K48- and K63-linked chains, consistent with previous observations (*Schaefer and Morgan, 2011*). Thus, it is not Ubp12 but the substrate that behaves unexpectedly. Notably, the ubiquitin signals that accumulate in Fzo1, Ubp2, Rad23 and Gpa1 are all composed of a limited number of discrete bands, instead of the high molecular weight smear, typical for polyubiquitylated substrates. For Fzo1, we find that Ubp12 recognizes ubiquitylated forms that only contain two ubiquitin moieties that are linked *via* K48. We propose that the presence of a short number of ubiquitin molecules on the ubiquitin chains recognized by Ubp12 could explain why they do not serve as a good signal for proteasomal degradation. The protein Met4 was also shown to be ubiquitylated with a a limited number of discrete bands (*Flick et al., 2004*; *Kuras et al., 2002*). In this case, intramolecular association with a ubiquitin binding domain in Met4 shields the ubiquitin chains, thus preventing their elongation and protecting Met4 against proteasomal degradation (*Flick et al., 2006*; *Tyrrell et al., 2010*).

## Regulation of DUB activity by ubiquitin

How deubiquitylation is controlled is poorly understood. Our findings suggest that this involves ubiquitylation of the DUBs themselves, because both Ubp2 and Ubp12 are regulated by ubiquitylation. This consequently renders DUBs interdependent, as exemplified with Ubp12 being the DUB of Ubp2. Interestingly, several examples in the literature illustrate a big diversity of DUB regulation (*Michel et al., 2017*). Therefore, additional mechanisms to proteolysis for the atypical function of Ubp2 ubiquitylation can be proposed. For example, Ubp2 ubiquitylation could induce a conformational change favouring catalytic activity, as observed for the DUB ATXN3 (*Todi et al., 2010*). This is supported by the observation that Ubp2 is among the largest yeast DUBs. In addition, several residues of Ubp2 were found to be phosphorylated (*Swaney et al., 2013*), suggesting that coordinated ubiquitylation/phosphorylation events could increase its activity. Finally, given that many DUBs often act as part of protein complexes, Ubp2 ubiquitylation could favor its interaction with Ubp12 and/or Cdc48. This could release autoinhibition by a conformational change, as observed for the DUB Ubp6 upon binding to the proteasome, i.e. a AAA+ ATPase like Cdc48 (*Hanna et al., 2006*). In fact, Cdc48 has been shown to associate with several DUBs (*Ossareh-Nazari et al., 2010*; *Papadopoulos et al., 2017*; *Rumpf and Jentsch, 2006*; *Uchiyama et al., 2002*) but also recognizes ubiquitylated proteins, consistent with its interaction with both Ubp12 and Ubp2. Therefore, DUB

ubiquitylation could allow recruitment of Cdc48 and provide a platform guiding DUBs to their relevant substrates. This would also justify the need for Fzo1-Cdc48 physical interaction. In fact, a local regulation of Fzo1 by Cdc48 could allow increased efficiency of the Cdc48-DUB cascade on Fzo1 regulation.

In conclusion, our results suggest that Cdc48 serves as a binding platform allowing cross-talk regulation between DUBs, bringing new insights into the knowledge of ubiquitin biology. These general findings open new perspectives to address some poorly understood questions, *e.g.* how Cdc48 regulates homotypic fusion events and how DUBs are interdependently regulated, possibly accounting for the multitude of DUBs present in a cell.

# Materials and methods

## Key resources table

| Reagent type (species) or resource | Designation | Source or reference | Identifiers | Additional information |
|---|---|---|---|---|
| strain (*Saccharomyces cerevisiae*) | Δfzo1 | PMID: 9483801 | Escobar_lab_stock_number: FA2 | |
| strain (*S. cerevisiae*) | cdc48-1 | PMID: 21441928 | Escobar_lab_stock_number: FA230 | |
| strain (*S. cerevisiae*) | cdc48-2 | PMID: 21441928 | Escobar_lab_stock_number: FA231 | |
| strain (*S. cerevisiae*) | cdc48-3 | PMID: 21441928 | Escobar_lab_stock_number: FA232 | |
| strain (*S. cerevisiae*) | Δubp2 | PMID: 9483801 | Escobar_lab_stock_number: FA260 | |
| strain (*S. cerevisiae*) | Δubp12 | PMID: 9483801 | Escobar_lab_stock_number: FA269 | |
| strain (*S. cerevisiae*) | Δfzo1 Δubp2 | PMID: 23317502 | Escobar_lab_stock_number: FA362 | |
| strain (*S. cerevisiae*) | Δubp2 Δubp12 | PMID: 23317502 | Escobar_lab_stock_number: FA382 | |
| strain (*S. cerevisiae*) | Δubp12 Δmdm30 | this study | Escobar_lab_stock_number: FA390 | *UBP12*::kanMX4; *MDM30*::kanMX4;obtained by crossing |
| strain (*S. cerevisiae*) | HA-Fzo1$^{int}$ in wt | PMID: 23317502 | Escobar_lab_stock_number: FA407 | |
| strain (*S. cerevisiae*) | HA-Fzo1$^{int}$ in Δubp2 | PMID: 23317502 | Escobar_lab_stock_number: FA415 | |
| strain (*S. cerevisiae*) | HA-Fzo1$^{int}$ in Δubp2 Δmdm30 | PMID: 23317502 | Escobar_lab_stock_number: FA427 | |
| strain (*S. cerevisiae*) | Δfzo1 Δubp12 | this study | Escobar_lab_stock_number: FA432 | *FZO1*::kanMX4; *UBP12*::kanMX4;obtained by crossing |
| strain (*S. cerevisiae*) | HA-Fzo1-K464R$^{int}$ in wt | this study | Escobar_lab_stock_number: FA451 | HA-Fzo1$^{K464R}$ genomically integrated with NatNT2 into RS140 |
| strain (*S. cerevisiae*) | wt (BY4741) | PMID: 9483801 | Escobar_lab_stock_number: RS140 | |
| strain (*S. cerevisiae*) | cdc48-2 Δfzo1 | this study | Escobar_lab_stock_number: RS430 | *FZO1*::natNT2 in FA231 |
| strain (*S. cerevisiae*) | cdc48-2 Δubp12 | this study | Escobar_lab_stock_number: RS466 | *FZO1*::hphNT1 in FA231 |
| strain (*S. cerevisiae*) | cdc48-2 Δubp2 Δubp12 | this study | Escobar_lab_stock_number: RS499 | *UBP12*::natNT2; UBP2::hphNT1 in FA231 |
| strain (*S. cerevisiae*) | Δdoa1 | PMID: 9483801 | Escobar_lab_stock_number: RS518 | |
| strain (*S. cerevisiae*) | Δpdr5 Δsnq2 | other | Escobar_lab_stock_number: RS527 | gift by J. Dohmen (YGA58): MATa, ADE2 his3-D200 leu2-3,112 lys2-801, trp1D63 ura3-52 *PDR5*::hphNT1 *SNQ2*::kanMX4 |
| strain (*S. cerevisiae*) | Ubp12-Flag$^{int}$ in cdc48-2 | this study | Escobar_lab_stock_number: RS546 | Ubp12-Flag genomically integrated with NatNT2 into FA231 |
| strain (*S. cerevisiae*) | Ubp12-Flag$^{int}$ in wt | this study | Escobar_lab_stock_number: RS547 | Ubp12-Flag genomically integrated with NatNT2 into BY4741 |
| strain (*S. cerevisiae*) | Δpdr5 Δsnq2 | this study | Escobar_lab_stock_number: RS554 | *PDR5*::NatNT2; SNQ2::hphNT1 in RS140 |
| strain (*S. cerevisiae*) | Δfzo1 Δdnm1 Δubp12 | this study | Escobar_lab_stock_number: RS556 | UBP12::NatNT2 in TS1028 |

*Continued on next page*

*Continued*

| Reagent type (species) or resource | Designation | Source or reference | Identifiers | Additional information |
|---|---|---|---|---|
| strain (*S. cerevisiae*) | Δpdr5 Δsnq2 cdc48-2 | this study | Escobar_lab_stock_number: RS559 | *PDR5*::NatNT2; *SNQ2*:: hphNT1 in FA231 |
| strain (*S. cerevisiae*) | cdc48-2 Δubp2 | this study | Escobar_lab_stock_number: TS686 | *UBP2*::hphNT1 in FA231 |
| strain (*S. cerevisiae*) | Δfzo1 Δdnm1 | other | Escobar_lab_stock_number: TS1028 | gift by B. Westermann (*SB95*): *FZO1*::kanMX4; *DNM1*:: kanMX4; obtained by crossing |
| strain (*S. cerevisiae*) | wt (DF5) | PMID: 11007476 | Escobar_lab_stock_number: TS1124 | |
| strain (*S. cerevisiae*) | ufd1-2 | PMID: 11847109 | Escobar_lab_stock_number: TS1125 | |
| strain (*S. cerevisiae*) | npl4-1 | PMID: 8930904 | Escobar_lab_stock_number: TS1126 | |
| strain (*S. cerevisiae*) | Ubp2-9Mycint in wt | this study | Escobar_lab_stock_number: TS1134 | Ubp2-9Myc genomically integrated with NatNT2 into RS140 |
| strain (*S. cerevisiae*) | Ubp2-3HAint in wt | this study | Escobar_lab_stock_number: TS1144 | Ubp2-3HA genomically integrated with hphNT1 in RS140 |
| strain (*S. cerevisiae*) | Ubp2-3HAint in Δubp12 | this study | Escobar_lab_stock_number: TS1147 | Ubp2-3HA genomically integrated with hphNT1 in FA269 |
| strain (*S. cerevisiae*) | pGAL-Ubp12-Flagint in wt | this study | Escobar_lab_stock_number: TS1153 | pGAL-Ubp12-Flag genomically integratedwith kanMX4 into RS544 |
| recombinant DNA reagent | pRS316 (plasmid) | PMID: 2659436 | Escobar_lab_stock_number: p8 | |
| recombinant DNA reagent | HA-Fzo1 on pRS316 (plasmid) | PMID: 23317502 | Escobar_lab_stock_number: p10 | |
| recombinant DNA reagent | HA-Fzo1-K464R on pRS316 (plasmid) | PMID: 23317502 | Escobar_lab_stock_number: p14 | |
| recombinant DNA reagent | YEplac181 (plasmid) | PMID: 3073106 | Escobar_lab_stock_number: p58 | |
| recombinant DNA reagent | Ubp2-Flag on YEplac181(plasmid) | PMID: 23317502 | Escobar_lab_stock_number: p59 | |
| recombinant DNA reagent | Ubp2-C745S-Flag on YEplac181(plasmid) | PMID: 23317502 | Escobar_lab_stock_number: p60 | |
| recombinant DNA reagent | Ubp12-Flag on YEplac181(plasmid) | PMID: 23317502 | Escobar_lab_stock_number: p61 | |
| recombinant DNA reagent | Ubp12-C372S-Flag on YEplac181(plasmid) | PMID: 23317502 | Escobar_lab_stock_number: p62 | |
| recombinant DNA reagent | YEplac195 (plasmid) | PMID: 3073106 | Escobar_lab_stock_number: p63 | |
| recombinant DNA reagent | Ubp12$^{C372S}$ on YEplac195 (plasmid) | this study | Escobar_lab_stock_number: p65 | Ubp12$^{C372S}$ (non-tagged) on YEplac195, 2μ, Ura3 |
| recombinant DNA reagent | mt-GFP on pYX142 (plasmid) | PMID: 11054823 | Escobar_lab_stock_number: p70 | |
| recombinant DNA reagent | Cdc48 wt on pRS313 (plasmid) | PMID: 22580068 | Escobar_lab_stock_number: p75 | |
| recombinant DNA reagent | pRS313 (plasmid) | PMID: 2659436 | Escobar_lab_stock_number: p79 | |
| recombinant DNA reagent | Cdc48-A547T on pRS313 (plasmid) | this study | Escobar_lab_stock_number: p150 | Cdc48$^{A547T}$ on pRS313, cen, His3 |
| recombinant DNA reagent | Ub on pKT10 (plasmid) | PMID: 2164637 | Escobar_lab_stock_number: p341 | |
| recombinant DNA reagent | Ub-K48R on pKT10 (plasmid) | PMID: 2164637 | Escobar_lab_stock_number: p342 | |
| recombinant DNA reagent | Ub-K63R on pKT10 (plasmid) | PMID: 2164637 | Escobar_lab_stock_number: p343 | |

*Continued on next page*

*Continued*

| Reagent type (species) or resource | Designation | Source or reference | Identifiers | Additional information |
|---|---|---|---|---|
| recombinant DNA reagent | Ub-K48R,K63R on pKT10 (plasmid) | PMID: 2164637 | Escobar_lab_stock_number: p344 | |
| recombinant DNA reagent | Myc-Ub on pRS426 (plasmid) | PMID: 25620559 | Escobar_lab_stock_number: p356 | |
| recombinant DNA reagent | pRS426 (plasmid) | PMID: 25620559 | Escobar_lab_stock_number: p375 | |
| Antibody | anti-Cdc48 | other | | gift by T. Sommer; (1:1,000/1:10,000) |
| Antibody | anti-Cox2 | other | | gift by W. Neupert; (1:5,000) |
| Antibody | anti-Flag M2 | Sigma | Sigma: F1804 | (1:1,000) |
| Antibody | anti-Fzo1 | this study | | Produced by GenScript using the peptide CHGDRKPDDDPYSSS; (1:1,000) |
| Antibody | anti-HA | Roche | Roche: 11867423001 | (1:1,000) |
| Antibody | anti-Myc | Cell Signaling | Cell_Signaling: #2276 | (1:1,000) |
| Antibody | anti-Sec61 | other | | gift by T. Sommer; (1:10,000) |
| Antibody | anti-Ssc1 | Fölsch et al., 1998 | | (1:40,000) |
| Antibody | anti-Tom40 | other | | gift by W. Neupert; (1:40,000) |
| Antibody | anti-Tpi1 | other | | gift by J. Dohmen; (1:5,000) |
| Antibody | anti-Ub (P4D1) | Cell Signaling | Cell_Signaling: #3936 | (1:1,000) |
| Antibody | anti-Ubc6 | other | | gift by T. Sommer; (1:10,000) |
| Antibody | anti-Ubp12 | this study | | (1:200) |
| software | Microsoft Office 2010 | Micosoft Corporation | | |
| software | Adobe Photoshop CS6 | Adobe | | |
| software | Adobe Illustrator CS6 | Adobe | | |
| software | Clone Manager | Sci-Ed Software | | |
| software | Image Quant | GE Healthcare Life Sciences | | |
| software | Axiovision | Zeiss | | |
| software | StepOne System | Thermo Fisher Scientific | | |
| kit | NucleoSpin RNA | Machery Nagel | REF:740955 | |
| kit | SuperScript III First-Strand Synthesis System | Invitrogen | Catalogue_number: 18080051 | |

## Yeast strains and growth media

See *Table 1* for details of all yeast strains used. Except for Δ*pdr5* Δ*snq2* (YGA58, from J. Dohmen) and *ufd1-2*, *npl4-1* and their corresponding wild type (DF5, from S. Jentsch) all other yeast strains are isogenic to the S288c (Euroscarf). They were grown according to standard procedures to the exponential growth phase at 30°C (unless stated otherwise) on complete (YP) or synthetic (SC) media supplemented with 2% (w/v) glucose (D), 2% (w/v) galactose or 2% (w/v) lactate (Lac). Cycloheximide (CHX) (Sigma, Germany) (100 µg/ml for protein shut-down, or 0.5 µg/ml when indicated, from a stock of 10 mg/ml in $H_2O$) or MG132 (Calbiochem) (50 or 100 µM from a stock of 10 mM in DMSO) was added when indicated.

## Plasmids

All plasmids used in this study are described in *Table 2*. Plasmid #65, encoding a non-tagged Ubp12$^{C372S}$ variant, expressed under the control of the *ADH1* promoter, was amplified from

**Table 1.** Yeast strains used in this study.

| Strain # | Strain name | Genotype | Reference |
|---|---|---|---|
| FA2 | Δfzo1 | FZO1::kanMX4 in BY4741 | *Brachmann et al., 1998* |
| FA230 | cdc48-1 | cdc48-1::KanMX4 in BY4741 | *Li et al. (2011)* |
| FA231 | cdc48-2 | cdc48-2::KanMX4 in BY4741 | *Li et al. (2011)* |
| FA232 | cdc48-3 | cdc48-3::KanMX4 in BY4741 | *Li et al. (2011)* |
| FA260 | Δubp2 | UBP2::kanMX4 in BY4741 | *Brachmann et al., 1998* |
| FA269 | Δubp12 | UBP12::kanMX4 in BY4741 | *Brachmann et al., 1998* |
| FA362 | Δfzo1 Δubp2 | FZO1::kanMX4; UBP2::kanMX4; obtained by crossing | *Anton et al. (2013)* |
| FA382 | Δubp2 Δubp12 | UBP12::kanMX4; UBP2::kanMX4; obtained by crossing | *Anton et al. (2013)* |
| FA390 | Δubp12 Δmdm30 | UBP12::kanMX4; MDM30::kanMX4; obtained by crossing | this study |
| FA407 | HA-Fzo1int in wt | HA-Fzo1 genomically integrated with NatNT2 into RS140 | *Anton et al. (2013)* |
| FA415 | HA-Fzo1int in Δubp2 | HA-Fzo1 genomically integrated with NatNT2 into FA260 | *Anton et al. (2013)* |
| FA427 | HA-Fzo1int in Δubp2 Δmdm30 | HA-Fzo1 genomically integrated with NatNT2 into Δubp2 Δmdm30 | *Anton et al. (2013)* |
| FA432 | Δfzo1 Δubp12 | FZO1::kanMX4; UBP12::kanMX4; obtained by crossing | this study |
| FA451 | HA-Fzo1-K464Rint in wt | HA-Fzo1$^{K464R}$ genomically integrated with NatNT2 into RS140 | this study |
| RS140 | wt | BY4741; S288C isogenic yeast strain; MATa, his3Δ1, leu2Δ0, met15Δ0, ura3Δ0 | *Brachmann et al., 1998* |
| RS430 | cdc48-2 Δfzo1 | FZO1::natNT2 in FA231 | this study |
| RS466 | cdc48-2 Δubp12 | FZO1::hphNT1 in FA231 | this study |
| RS499 | cdc48-2 Δubp2 Δubp12 | UBP12::natNT2; UBP2::hphNT1 in FA231 | this study |
| RS518 | Δdoa1 | DOA1::kanMX4 in BY4741 | *Brachmann et al., 1998* |
| RS527 | Δpdr5 Δsnq2 | MATa, ADE2 his3-D200 leu2-3,112 lys2-801, trp1D63 ura3-52 PDR5::hphNT1 SNQ2::kanMX4 | J. Dohmen (YGA58) |
| RS546 | Ubp12-Flagint in cdc48-2 | Ubp12-Flag genomically integrated with NatNT2 into FA231 | this study |
| RS547 | Ubp12-Flagint in wt | Ubp12-Flag genomically integrated with NatNT2 into BY4741 | this study |
| RS554 | Δpdr5 Δsnq2 | PDR5::NatNT2; SNQ2::hphNT1 in RS140 | this study |
| RS556 | Δfzo1 Δdnm1 Δubp12 | UBP12::NatNT2 in TS1029 | this study |
| RS559 | Δpdr5 Δsnq2 cdc48-2 | PDR5::NatNT2; SNQ2::hphNT1 in FA231 | this study |
| TS686 | cdc48-2 Δubp2 | UBP2::hphNT1 in FA231 | this study |
| TS1029 | Δfzo1 Δdnm1 | FZO1::kanMX4; DNM1::kanMX4; Mat α, BY background, obtained by crossing | B. Westermann (#94) |
| TS1124 | wt (DF5) | MATα, trp1-1(am), ura3-52, his3Δ200, leu2-3, lys2-801 | *Hoppe et al. (2000)* |
| TS1125 | ufd1-2 | ufd1-2$^{ts}$ in TS1124 | *Braun et al. (2002)* |
| TS1126 | npl4-1 | npl4-1$^{ts}$ in TS1124 | *DeHoratius and Silver (1996)* |
| TS1134 | Ubp2-9Mycint in wt | Ubp2-9Myc genomically integrated with NatNT2 into RS140 | this study |
| TS1144 | Ubp2-3HAint in wt | Ubp2-3HA genomically integrated with hphNT1 in RS140 | this study |
| TS1147 | Ubp2-3HAint in Δubp12 | Ubp2-3HA genomically integrated with hphNT1 in FA269 | this study |
| TS1153 | pGAL-Ubp12-Flagint in wt | pGAL-Ubp12-Flag genomically integrated with kanMX4 into RS544 | this study |

DOI: https://doi.org/10.7554/eLife.30015.022

Ubp12$^{C372S}$-Flag and cloned with Pst1, Sal1 into the same sites of YEplac195. Plasmid #150, encoding Cdc48$^{A547T}$ was generated by point mutagenesis using plasmid #75.

## Antibodies

All antibodies used in this study are described in *Table 3*.

**Table 2.** Plasmids used in this study.

| Plasmid # | Plasmid name | Description | Bacterial selection | Reference |
|---|---|---|---|---|
| 8 | pRS316 | pRS316, cen, Ura3 | Amp | *Sikorski and Hieter, 1989* |
| 10 | HA-Fzo1 on pRS316 | HA-Fzo1 on pRS316, Fzo1 prom, cen, Ura3 | Amp | *Anton et al. (2013)* |
| 14 | HA-Fzo1-K464R on pRS316 | HA-Fzo1$^{K464R}$ on pRS316, Fzo1 prom, cen, Ura3 | Amp | *Anton et al. (2013)* |
| 58 | YEplac181 | YEplac181, 2μ, Leu2 | Amp | *Gietz and Sugino, 1988* |
| 59 | Ubp2-Flag on YEplac181 | Ubp2-Flag on YEplac181, Adh1 prom, 2μ, Leu2 | Amp | *Anton et al. (2013)* |
| 60 | Ubp2-C745S-Flag on YEplac181 | Ubp2$^{C745S}$-Flag on YEplac181, Adh1 prom, 2μ, Leu2 | Amp | *Anton et al. (2013)* |
| 61 | Ubp12-Flag on YEplac181 | Ubp2-Flag on YEplac181, Adh1 prom, 2μ, Leu2 | Amp | *Anton et al. (2013)* |
| 62 | Ubp12-C372S-Flag on YEplac181 | Ubp2$^{C372S}$-Flag on YEplac181, Adh1 prom, 2μ, Leu2 | Amp | *Anton et al. (2013)* |
| 63 | YEplac195 | YEplac195, 2μ, Ura3 | Amp | *Gietz and Sugino, 1988* |
| 65 | Ubp12$^{C372S}$ on YEplac195 | Ubp12$^{C372S}$ (non-tagged) on YEplac195, 2μ, Ura3 | Amp | this study |
| 70 | mt-GFP on pYX142 | mt-GFP on pYX142, cen, Leu2 | Amp | *Westermann and Neupert, 2000* |
| 75 | Cdc48 wt on pRS313 | Cdc48 wt on pRS313, cen, His3 | Amp | *Esaki and Ogura (2012)* |
| 79 | pRS313 | pRS313, cen, His3 | Amp | *Sikorski and Hieter, 1989* |
| 150 | Cdc48-A547T on pRS313 | Cdc48$^{A547T}$ on pRS313, cen, His3 | Amp | this study |
| 341 | Ub on pKT10 | Ub on pK10, 2μ, Ura3 | Amp | *Tanaka et al., 1990* |
| 342 | Ub-K48R on pKT10 | Ub$^{K48R}$ on pK10, 2μ, Ura3 | Amp | *Tanaka et al., 1990* |
| 343 | Ub-K63R on pKT10 | Ub$^{K63R}$ on pK10, 2μ, Ura3 | Amp | *Tanaka et al., 1990* |
| 344 | Ub-K48R,K63R on pKT10 | Ub$^{K48R,K63R}$ on pK10, 2μ, Ura3 | Amp | *Tanaka et al., 1990* |
| 356 | Myc-Ub on pRS426 | pCup1-Myc-Ub on pRS426, 2μ, Ura3 | Amp | *Li et al., 2015* |
| 375 | pRS426 | pRS426, 2μ, Ura3 | Amp | *Li et al., 2015* |

DOI: https://doi.org/10.7554/eLife.30015.023

## Spot tests

For growth assays, serial 1:5 dilutions of exponentially growing cells using a starting OD$_{600}$ of 0.5 or 0.005 were spotted on YP or SC media containing glucose or lactate and were grown at 30°C or 37°C, as indicated.

**Table 3.** Antibodies used in this study.

| Name | Dilution | Reference |
|---|---|---|
| Cdc48 | 1:1000/1:10,000 | T. Sommer |
| Cox2 | 1:5000 | W. Neupert |
| Flag M2 | 1:1000 | Sigma (F1804) |
| Fzo1 | 1:1000 | this study |
| HA | 1:1000 | Roche (11867423001) |
| Myc | 1:1000 | Cell Signaling (#2276) |
| Sec61 | 1:10,000 | T. Sommer |
| Ssc1 | 1:40,000 | *Fölsch et al., 1998* |
| Tom40 | 1:40,000 | W. Neupert |
| Tpi1 | 1:5000 | J. Dohmen |
| Ub (P4D1) | 1:1000 | Cell Signaling (#3936) |
| Ubc6 | 1:10,000 | T. Sommer |
| Ubp12 | 1:200 | this study |

DOI: https://doi.org/10.7554/eLife.30015.024

## Protein steady state levels and synthesis shutoff

For analysis of protein steady state levels, total proteins from 3 $OD_{600}$ exponentially growing cells were extracted at alkaline pH (*Escobar-Henriques et al., 2006*) and analyzed by SDS-PAGE and immunoblotting. To monitor protein turnover, cycloheximide (100 µg/ml) was added to exponential cells. Samples of 3 $OD_{600}$ cells were collected at the indicated time points and total proteins were extracted and analyzed as described above. For monitoring proteasome-dependent degradation of endogenous Fzo1 in wt and *cdc48-2* cells, additionally deleted for *SNQ2* and *PDR5*, YPD media was used (*Liu et al., 2007*), and cells were treated with 50 µM MG132, 30 min before adding cycloheximide. For monitoring proteasome-dependent degradation of Ubp2, expressed from plasmid #59, SCD media was used, and 50 µM MG132 was added 1 hr before starting the cycloheximide chase. Western blots were quantified using Image Quant (GE Healthcare, Illinois, USA). Levels of the protein of interest at time zero were set to 1. Mean values are shown and the error bars reflect the standard deviation (SD).

## Analysis of free ubiquitin and ubiquitin-conjugates

Total proteins were extracted as described above for the analysis of protein steady state levels but solubilized in LDS buffer (Thermo Fisher Scientific, Massachusetts, USA). Samples were run on precast 4–12% bis-tris gels (Thermo Fisher Scientific) using MES buffer (50 mM MES, 50 mM Tris Base, 0.1% SDS, 1 mM EDTA, pH 7.3) and transferred to PVDF membranes. Membranes were treated with denaturing solution (6 M guanidium chloride, 20 mM Tris pH 7.5, 1 mM PMSF, 5 mM β-mercaptoethanol) for 30 min and then washed before blocking. Proteins were detected with a ubiquitin-specific antibody (P4D1; Cell Signaling, Massachusetts) and a Tpi1-specific antibody, as a loading control. Quantifications were performed using Image Quant (GE Healthcare). Wt values were set to one and the mutants are shown in relation to the wt. Mean values are shown and the error bars reflect the standard deviation (SD).

## Analysis of Ubp12 ubiquitylation

Immunoprecipitation of Ubp12$^{C372S}$-Flag was performed as follows: 160 $OD_{600}$ of yeast cells grown in SCD media to the exponential growth phase were disrupted with glass beads (0.4–0.6 µm) in TBS. After centrifugation, at 16000 g for 10 min, the supernatant was employed to perform an overnight precipitation of Ubp12$^{C372S}$-Flag, using Flag-coupled beads (Sigma-Aldrich). Elution was performed for 2 hr shaking at 4°C with the 3xFlag-peptide (Sigma; 200 µg/ml final concentration) in the following buffer: 50 mM Tris-HCl pH 7.5, 50 mM NaCl. After adding Laemmli buffer, the eluate was split in two, proteins were then resolved in 7% Tris-acetate gels as described (*Cubillos-Rojas et al., 2012*). After transfer, the nitrocellulose membrane was divided in two: one half was immunoblotted with a Flag-specific (Sigma) and the other half with a ubiquitin-specific antibody (P4D1; Cell Signaling).

## Analysis of Ubp2 ubiquitylation

Immunoprecipitation of Ubp2$^{C745S}$-Flag was performed as follows: 160 $OD_{600}$ of yeast cells grown in SCD media to the exponential growth phase were disrupted with glass beads (0.4–0.6 µm) in RIPA buffer without detergents (HEPES-KOH 40 mM pH 7.6, NaCl 150 mM, EDTA 5 mM). After centrifugating at 16000 g for 10 min, the supernatant was diluted in an equal volume of RIPA buffer containing 2X detergents, so that the final composition was HEPES-KOH 40 mM pH 7.6, NaCl 150 mM, EDTA 5 mM, Triton X100 1%, SDS 0.1%, sodium deoxycholate 0.5%. After sonication for 15 min at 4°C in a water bath, denatured cytosolic fractions were employed to precipitate Ubp2$^{C745S}$-Flag. Flag-coupled beads (Sigma-Aldrich) were used for overnight immunoprecipitation and protein elution was performed with Laemmli buffer for 20 min shaking at 40°C. The eluate was split in two and resolved in 8% Tris-glycine gels. After transfer, the nitrocellulose membrane was divided in two: one half of the eluate was immunoblotted with a Flag-specific (Sigma) and the other half with a ubiquitin-specific antibody (P4D1; Cell Signaling).

## Analysis of Fzo1 ubiquitylation

Fzo1 ubiquitylation was analyzed as follows: 160 $OD_{600}$ cell pellets of exponentially growing cultures were used to obtain crude mitochondrial extracts as described (*Anton et al., 2013*). After solubilization with 0.2 % NG310 (Lauryl Maltose Neopentyl Glycol; Anatrace) for 1 hr rotating at 4°C, samples

were centrifuged and 10% of the supernatant was kept as input material. After denaturing in Laemmli buffer for 20 min shaking at 40℃ samples were resolved by SDS-PAGE. If necessary, the remaining 90% of the supernatant was incubated with HA-coupled beads (Sigma-Aldrich) overnight rotating at 4℃. Three washes were performed with 0.2 % NG310 in TBS. HA-Fzo1 was eluted in 50 μl of Laemmli buffer for 20 min shaking at 40℃ and analyzed by SDS-PAGE. Proteins were transferred onto nitrocellulose membranes and subsequently immunoblotted using an HA-specific antibody (Roche, Switzerland).

## Co-immunoprecipitations

### Interaction between Ubp12-Flag and Cdc48

160 $OD_{600}$ of yeast cells grown in complete media to the exponential growth phase were disrupted with glass beads (0.4–0.6 μm) in TBS. After centrifugation at 16000 g for 10 min, the crude membrane fraction was solubilized using 0.5% digitonin for 1 hr rotating at 4℃. $Ubp12^{C372S}$-Flag was immunoprecipitated using Flag-coupled beads (Sigma-Aldrich) for 2 hr rotating at 4℃. Beads were washed three times with 0.1% digitonin in TBS and $Ubp12^{C372S}$-Flag was eluted in Laemmli buffer for 20 min shaking at 40℃. 10% of the input and 100% of the eluate fractions were analyzed by SDS-PAGE and immunoblotting using Flag-specific (Sigma) and Cdc48-specific antibodies.

### Interaction between HA-Fzo1 and Cdc48

Performed as described above for the Ubp12-Cdc48 interaction, with the following modifications: solubilization was performed with 0.2 % NG310; immunoprecipitation was performed for 2 hr using HA-coupled beads (Sigma-Aldrich) pre-blocked with PVPK30 (Polyvinylpyrrolidone; Fluka); washes were performed with 0.2 % NG310 in TBS. 4% of the input and 50% of the eluate fractions were analyzed by SDS-PAGE and immunoblotting using HA-specific (Roche) and Cdc48-specific antibodies.

### Interaction between Ubp2-Flag and Ubp12

Immunoprecipitation of $Ubp12^{C372S}$ was performed as follows: 160 $OD_{600}$ of yeast cells grown in SCD media to the exponential growth phase were disrupted with glass beads (0.4–0.6 μm) in TBS. After centrifugation at 16000 g for 10 min, the cytosolic fraction was used to precipitate $Ubp12^{C372S}$ by using an Ubp12-specific antibody and the affinity resin with protein G immobilized (Protein G Sepharose 4 Fast Flow; GE Healthcare). After 3 hr rotating at 4℃, beads were washed three times in TBS. Protein elution was performed with Laemmli buffer for 20 min shaking at 40℃. 1% of the input and 100% of the eluate were analyzed by SDS-PAGE and immunoblotting using Flag- and Ubp12-specific antibodies.

## Mitochondrial morphology

Yeast strains were transformed with mitochondrial-targeted GFP, grown on YPD or SC media to the exponential phase and analyzed as described (*Escobar-Henriques et al., 2006*) by epifluorescence microscopy (Axioplan 2; Carl Zeiss MicroImaging, Inc., Germany) using a 100x oil-immersion objective. Images were acquired with a camera (AxioCam MRm, Carl Zeiss MicroImaging, Inc.) and processed with Axiovision 4.7 (Carl Zeiss MicroImaging, Inc.).

## Analysis of mtDNA content using RT-PCR

RNA was isolated from 2 $OD_{600}$ exponentially growing yeast cells using the NucleoSpin RNA kit (Macherey Nagel, Germany). cDNA was synthesized using the SuperScript III First-Strand Synthesis System (Invitrogen, Massachusetts, USA). mtDNA was quantified by the amplification of *COX3* and normalized to *ACT1* (as housekeeping gene). Essentially, a dilution of 1:100 of the cDNA was used for the amplification of *COX3* (fw: TTGAAGCTGTACAACCTACC; rv: CCTGCGATTAAGGCATGATG) and *ACT1* (fw: CACCCTGTTCTTTTGACTGA; rv: CGTAGAAGGCTGGAACGTTG) by RT-PCR using the Power SYBR Green Master Mix (AppliedBioSystems) and three technical replicates for each of the six biological replicates. The ΔCT was calculated using the Livak/$2^{-\Delta\Delta CT}$ method (*Livak and Schmittgen, 2001*) and the fold change of *COX3* RNA content in Δfzo1 and cdc48-2 was calculated in relation to wt.

## DUB assay

In vitro deubiquitylation assays were performed as described (*Hospenthal et al., 2015*), Essentially, purified K48 or K63 multi-Ub (BostonBiochem) or di-Ub chains (kindly gifted by Thomas Hermanns) were treated with the DUBs USP2 (BostonBiochem), USP21 (kindly gifted by Selver Altin) or Ubp12. Ubp12 was purified as described above, for the analysis of Ubp12 ubiquitylation, but glycerol to the final concentration of 10% was added, instead of Laemmli. Aliquots of 18 µl, corresponding to 80 $OD_{600}$ yeast cells, were frozen in liquid nitrogen and stocked at $-80°C$ until further use. For the DUB assay, per reaction, one aliquot of purified Ubp12-flag, 3 µM USP2 or 5 µM USP21 were pre-incubated with 1x DUB dilution buffer (25 mM Tris pH 7.5, 10 mM DTT, 150 mM NaCl) for 10 min at RT.

After pre-incubation, the DUBs were mixed with di- or multi-Ub chains to a final concentration of 5 µM in 1x DUB buffer (10x DUB buffer: 500 mMTris pH 7.5, 500 mMNaCl, 50 mM DTT). Different incubation conditions were used: Ubp12 was incubated with the Ub chains for 45 min at 30°C, USP2 and USP21 for 30 min at 37°C. The reactions were stopped by adding 4x Laemmli buffer. These mixtures were incubated for 20 min at 40°C shaking and further run on an 11% Tris-Tricine SDS-PAGE and transferred onto a PVDF membrane. Ponceau S was used to stain the membrane and after destaining with methanol for 5 min, the membrane was incubated in denaturing solution (6M guanidium chloride, 20 mMTris pH 7.5, 1 mM PMSF, 5 mMβ-mercaptoethanol) for 30 min. Extensive washing was done in TBS-T before blocking the membrane over night with 5% milk in TBS. Results were analyzed by immunoblotting using a Ub-specific antibody.

## Acknowledgements

We would like to thank T Sommer for the Cdc48 and Ubc6 antibodies, B Westermann for the plasmid pYX142-mtGFP and for the strain Δ*fzo1* Δ*dnm1*, J Dohmen for the Tpi1 antibody, for the strain YGA58 and for stimulating discussions, K Tanaka for the Ubiquitin plasmid and corresponding mutant variants, T Ogura for the Cdc48 plasmid, K Hofmann for stimulating discussions, S Altin, T Hermanns and K Hofmann for the USP21 enzyme and the purified di-ubiquitin chains and M Hospenthal and D Komander for help with the DUB assay protocol. We are grateful towards T Langer for critical input and support, the Langer lab for stimulating discussions, the Hoppe lab, especially W Pokrzywa, for technical advises and towards T Langer, T Hoppe and especially T Tatsuta for critical reading of the manuscript. This work was supported by grants of the Deutsche Forschungsgemeinschaft (DFG; ES338/3-1, SFB635 to ME-H), was funded under the Institutional Strategy of the University of Cologne within the German Excellence Initiative and benefited from funds of the Faculty of Mathematics and Natural Sciences, attributed to ME-H.

## Additional information

### Funding

| Funder | Grant reference number | Author |
|---|---|---|
| Deutsche Forschungsgemeinschaft | ES338/3-1 | Mafalda Escobar-Henriques |
| Universität zu Köln | German Excellence Initiative and Faculty of Mathematics and Natural Sciences | Mafalda Escobar-Henriques |
| Deutsche Forschungsgemeinschaft | SFB635 | Mafalda Escobar-Henriques |
| Deutsche Forschungsgemeinschaft | CRC1218TPA03 | Mafalda Escobar-Henriques |

The funders had no role in study design, data collection and interpretation, or the decision to submit the work for publication.

### Author contributions

Tânia Simões, Ramona Schuster, Conceptualization, Data curation, Formal analysis, Investigation, Methodology, Writing—review and editing; Fabian den Brave, Writing—review and editing, Initial

observations; Mafalda Escobar-Henriques, Conceptualization, Data curation, Formal analysis, Supervision, Funding acquisition, Investigation, Methodology, Writing—original draft, Project administration, Writing—review and editing

### Author ORCIDs
Tânia Simões ⬥ http://orcid.org/0000-0002-5971-4935
Mafalda Escobar-Henriques ⬥ http://orcid.org/0000-0002-0879-3119

### Decision letter and Author response
Decision letter https://doi.org/10.7554/eLife.30015.027
Author response https://doi.org/10.7554/eLife.30015.028

## Additional files
### Supplementary files
• Transparent reporting form
DOI: https://doi.org/10.7554/eLife.30015.025

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
