## [Decision Letter]

Thank you for submitting your article "Cdc48 regulates a deubiquitylase cascade critical for ubiquitin homeostasis and mitochondrial fusion" for consideration by *eLife*. Your article has been reviewed by three peer reviewers, and the evaluation has been overseen by a Guest Reviewing Editor and Vivek Malhotra as the Senior Editor. The reviewers have opted to remain anonymous.

The reviewers have discussed the reviews with one another and the Reviewing Editor has drafted this decision to help you prepare a revised submission.

Summary:

This manuscript investigates the regulation of Fzo1 during mitochondrial fusion. The authors focus specifically on how the conserved AAA ATPase Cdc48 together with two deubiquitinating enzymes (DUBs), Ubp12 and Ubp2, control Fzo1 ubiquitination status and its activity in mitochondrial fusion. It is shown that the two DUBs not only control the activity of Fzo1 but also of each other. In particular, Ubp2 levels are controlled by Ubp12. On the other hand, Cdc48 regulates Ubp12 stability suggesting that a cascade of ubiquitination events finely tunes Fzo1 fusogenic activity.

Essential revisions:

While all three reviewers find the results potentially interesting the following points must be addressed to solidify the key conclusion of this study.

1) The main claims of this manuscript rely on the degradation of DUBs under strong overexpression. Thus, the evaluation of Ubp2 and Ubp12 half-lives should be re-analyzed and characterized under endogenous expression.

2) The negative regulation of Ubp2 by Ubp12 is a central point of model proposed but not fully characterized. In principle the removal of ubiquitin from a substrate should increase and not decrease its stability so the authors should clarify how Ubp2 is destabilized by Ubp12.

3) The statistical analysis of numerous data should be improved.

The comments of the individual reviewers are pasted below. You might want to address the additional concerns of the reviewers in case you can.

*Reviewer #1:*

The manuscript by Simões et al. entitled "Cdc48 regulates a deubiquitylase cascade critical for ubiquitin homeostasis and mitochondria fusion" studies the function of the AAA+ Atpase Cdc48 in mitochondrial fusion. Cdc48 is a ubiquitin selective segregase chaperone that assists proteasomal degradation of a myriad of substrates, including proteins localized to the mitochondrial outer membranes. It was previously shown that in Parkin-mediated mitophagy in mammalian cells, the Cdc48 homologue p97 can directly target mitofusin for degradation. This process is required for fragmentation of mitochondrion and subsequent mitophagy. In this study, the authors report a distinct function of Cdc48 in mitochondrial dynamics. In this case, the regulation is executed through two deubiquitynases (DUBs) that appear to form a cascade. The genetic evidence presented are nice and convincing, but there are some technical concerns on the biochemical part of the study. The major issue is that the study relies too much on using overexpressed DUBs as Cdc48 substrate. In addition, the integrity of the proposed model depends on some assumptions that are uncertain.

Specific points:

1) The conclusions of this study rely heavily on overexpression of tagged deubiquitylases as the substrate of Cdc48. Considering the reported role of Cdc48 in diverse protein quality control processes, the authors need to exclude the possibility that overexpressed Ubp12 and Ubp2 are short-lived because of a potential misfolding issue caused by protein overexpression. One way to address this is to tag these DUBs endogenously and analyze their stability in wild type and Cdc48 mutant cells by cycloheximide chase.

2) In Figure 2, the authors show an interaction between Cdc48 and Fzo1 that is dependent on Fzo1 ubiquitination. However, it is unclear what is the functional consequence of this interaction. Is it also unclear whether the interaction is direct or indirect. In the subsection “Cdc48 supports turnover of ubiquitylated Ubp12”, the conclusion that Cdc48 specifically interacts with pro-fusion ubiquitin is not supported by any data.

3) In Figure 3, the authors try to demonstrate that Ubp12 is an unstable protein that is degraded in a Cdc48 dependent manner. All the experiments were done with overexpressed Ubp12. The authors should analyze the stability of endogenous Ubp12. In addition, several experiments shown are incomplete (e.g. Figure 3, also Figure 5). These experiments should include a cdc48 mutant for comparison.

4) Figure 6 shows that the stability of Ubp2 is regulated by Ubp12. Once again, only overexpressed Ubp2 was studied. It is also puzzling why loss of Ubp12 can stabilize Ubp2, because DUB deficiency usually leads to increased ubiquitination and turnover. The quality of Figure 5 needs to be improved.

5) The authors argue that there are two forms of ubiquitinated Fzo1; one supports mitochondrial fusion while the other mitigates it, but the nature of the pro-fusion ubiquitination is unclear. The authors should consider analyzing the ubiquitin linkages on Fzo1 that is accumulated in Ubp12 mutant cells. This is an important issue because it is a bit hard to understand how Ubp12 can act on two different kinds of ubiquitin signals; the one on Fzo1 is incompetent for degradation, but the one on Ubp2 seems to influence its stability in a way that is counter-intuitive. Thus, in my opinion, to make the model convincing, the authors need to characterize the Ubp2-mediated deubiquitination reaction in more detail. What kind of linkage is preferred by Ubp12? Are Ubp2 and Fzo1 conjugated with the same type of ubiquitin chains preferred by Ubp12? Do these chains increase protein stability as oppose to target them for degradation?

6) "Fzo1 ubiquitination requires its lysine 464." If Fzo1 carries two types of ubiquitin chains, are they both attached to lysine 464? If yes, how are these processes coordinated? If not, where is the pro-fusion ubiquitin chain attached?

*Reviewer #2:*

The manuscript by Simoes et al. describes a novel function of the evolutionary highly conserved AAA-ATPase Cdc48, which is a potent modulator of neurodegenerative diseases. The authors convincingly demonstrate that Cdc48 controls mitochondrial fusion by a novel deubiquitylase (DUB) cascade at the level of the mitochondrial fusion factor Fzo1. Cdc48 is needed for mitochondrial fusion, and its mutation interferes with the formation of the mitochondrial network. Cdc48 and two DUBs control Fzo1 levels by discrete ubiquitylation patterns: A stable pro-fusion ubiquitylation pattern and an instable pro-degradation ubiquitylation pattern. The pro-fusion ubiquitylation pattern is removed by the DUB, Ubp12, thereby inhibiting mitochondrial fusion. The pro-degradation ubiquitylation pattern is removed by the DUB, Ubp2, thereby stabilizing Fzo1 and promoting mitochondrial fusion. The level of complexity is increased by the fact that Ubp12 is required for UPS-dependent degradation of Ubp2. Thus, high levels of Ubp12 interfere with mitochondrial fusion by two mechanisms: First, the removal of the pro-fusion ubiquitylation pattern from Fzo1, and second the induction of Ubp2 degradation, which results in the degradation of Fzo1. Therefore, controlling Ubp12 activities by Cdc48 enables the fine-tuning of Fzo1-dependent mitochondrial fusion.

This study is of outstanding interest for researchers working on mitochondrial dynamics and the UPS-dependent control of mitochondrial activities. Since the human homolog of Cdc48 is a modulator of human disease, this study is potentially also important for the elucidation of novel disease mechanisms. Publication of this very well written manuscript can be recommended after the authors address the concerns outlined below.

1) Data in Figure 5 suggest that a DUB, Ubp12, is required for degradation of Ubp2. The authors show that Ubp2 is ubiquitylated and stabilized in the absence of Ubp12. In the presence of Ubp12, the ubiquitylated form of Ubp2 cannot be found and Ubp2 is degraded by the proteasome. In other words, the authors propose a model in which deubiquitylation of Ubp2 by Ubp12 promotes its degradation by the proteasome. It is very surprising that removal of ubiquitin from a substrate, rather than its addition to it, triggers proteasome-dependent degradation. In my eyes, this requires further explanations, ideally with additional experiments. I consider this an essential point, because the negative regulation of Ubp2 by Ubp12 is a central point of the authors' model.

2) The term "ubiquitin homeostasis" should be removed from the title, as this has not been addressed in the paper.

3) The authors did not use any statistical analysis (like Student's t-test or ANOVA) to substantiate their numerous quantifications. Moreover, they used standard errors instead of standard deviations for error bars. Since standard errors underrepresent the variance of data the authors have to show all their quantitative data with standard deviation. Alternatively, they can keep standard errors as error bars but must then provide appropriate statistical analysis.

In particular, the effects shown in Figure 4, Figure 4—figure supplement 1 and Figure 7 are rather moderate and must be substantiated by a more rigorous statistical analysis. If these results turn out to be not significant, the associated statements and conclusions have to be toned down accordingly.

*Reviewer #3:*

In their work entitled “Cdc48 regulates a deubiquitylase cascade critical for ubiquitin homeostasis and mitochondrial fusion", Escabor-Henriques and coworkers investigate the mechanism by which the ATPase Cdc48 regulates Fzo1 mediated mitochondrial fusion. The major claims are that (1) Cdc48 action stabilizes so-called "pro-fusion" ubiquitylation of Fzo1 by promoting the degradation of the deubiquitylase Ubp12. (2) Concomitantly, reduced Ubc12 levels result in a stabilization of Ubp2 which in turn reduces "anti-fusion" ubiquitylation, contributing to Fzo1 stability.

The findings presented in this manuscript are certainly interesting as they give more mechanistic detail on the earlier finding by the same group that the two DUBs Ubp2 and Ubp12 regulate mitochondrial fusion (Anton et al. 2013), and provide an explanation for the observation by Esaki and Ogura (2012) that Cdc48 plays a role in mitochondrial fusion.

I do have some concerns though that should be addressed before publishing:

1) An important finding of this work is that the DUB Ubp12 is degraded in a Cdc48-dependent manner. Data for this is presented in Figure 3. In these experiments (and in others in which a DUB is expressed from a plasmid), Ubp12 is strongly overexpressed from a 2µ plasmid with the strong ADH promoter (as described in Anton et al., 2013). It is possible that degradation occurs because of the overexpression, e.g. because of then sub-stoichiometric amounts of a binding partner that normally stabilizes Ubp12. There are many examples in the literature for the degradation of orphan subunits in the literature, that are otherwise stable proteins (e.g. 1: Braun and Jentsch, 2007, EMBO Rep. 8(12):1176-82; 2: Habeck et al., 2015, JCB 209(2):261-73). I suggest that either this experiment is repeated using a chromosomally tagged version of Ubp12 under its endogenous promoter or that the possibility of an artifact due to overexpression is at least discussed. The latter would suffice in my opinion because the interplay of Cdc48 and Ubp12 is nicely shown in the following Figure 4 and its supplement.

2) In Figure 2, the authors provide evidence that Cdc48 is physically interacting with Fzo1, depending on ubiquitin(-chains) on K464. What is the proposed function of this interaction?

3) Cdc48 seems to have other functions in this pathway, apart from regulating Ubp12, since steady state levels of Fzo1 are not restored in cdc48-2/ ubc12Delta cells. Please comment.

4) Figure 4—figure supplement 1 are insufficiently explained. Especially panel D is difficult to understand since Dnm1 is not introduced and the rationale for the experiment is missing.

5) A table providing information on plasmids used in this study would be helpful, especially since information about expression of DUBs (overexpression, ADH promoter) is somewhat hidden by merely referencing Anton et al. (2013).

6) The discussion about a role of Cdc48 in membrane fusion remains rather superficial. The proposed mechanism for the role of ubiquitination in Syntaxin 5-mediated fusion is rather different, namely that it prevents SNARE pairing. Here, p97 would mediate deubiquitination of Syn5 and thereby activate Syntaxin 5 (Huang et al. 2016). Furthermore, there is otherwise little evidence for a "general role of Cdc48" in membrane fusion. I suggest rephrasing of this paragraph.

[Editors' note: further revisions were requested prior to acceptance, as described below.]

Thank you for resubmitting your work entitled "Cdc48 regulates a deubiquitylase cascade critical for mitochondrial fusion" for further consideration at *eLife*. Your revised article has been favorably evaluated by Vivek Malhotra (Senior Editor), a Guest Reviewing Editor, and three reviewers.

The manuscript has been improved but there are some remaining issues that need to be addressed before acceptance, as outlined below:

The concerns from Reviewer 1 should be addressed in full. In particular, given the small changes in the turnover of endogenous Ubp2 (Figure 5) it is important to test whether they are statistically significant. Moreover, the authors should discuss alternative mechanisms of Ubp2 regulation and that may be independent of its proteolysis.

*Reviewer #1:*

This is a revised manuscript that addresses the role of Cdc48, an AAA ATPase in mitochondria dynamics using yeast as a model. In the first-round review, a major problem identified by all referees is that the analyses of protein stability are based entirely on overexpressed Usp12 and Usp2. The authors now tagged Ubp12 and Ubp2 endogenously and analyzed their turnover in different genetic backgrounds. However, the newly collected data do not seem to support the authors’ main conclusions.

1) Subsection “Cdc48 supports turnover of ubiquitylated Ubp12” – the authors concluded that the stability of endogenous Ubp12 is regulated by Cdc48. However, the phenotype in my opinion is quite weak (Figure 3). Although the authors provide quantification results, the representative gel does not convince me that endogenous Ubp12 is unstable. Compared to Ubc6, a previously documented Cdc48 substrate, the accumulation of Ubp12 in cdc48-2 mutant cells is marginal, and the turnover is not as obvious as that of overexpressed Ubp12 (Figure 3—figure supplement 1), suggesting that overexpressed Ubp12 may not be properly folded and thus becomes a Cdc48 substrate. Given that there may be only a small increase in Ubp12 protein level in cdc48-2 mutant cells, I am not convinced that the role of Cdc48 in regulation of mitochondria dynamics is achieved through controlling the stability of Ubp12.

2) Another major conclusion of the study is that the two DUBs form a regulatory "cascade" with the stability of Ubp2 being controlled by Ubp12. The new result shown in Figure 5 does show that endogenous Ubp2 is unstable, but intriguingly, although the degradation of Ubp2 appears to be inhibited when Ubp12 gene was deleted, there is no obvious accumulation of Ubp2 in ubp12 deficient cells. Thus, it is unclear how Ubp12 could regulate the stability of Fzo1 in a Ubp2 dependent manner.

Other issues:

3) Subsection “Cdc48 promotes mitochondrial fusion and prevents Fzo1 turnover” – "Consistent with…". The authors conclude that the levels of Fzo1 were slightly decreased in the cdc48-3 mutant or in cells deleted for the Cdc48 co-factor factors Npl4, Ufd1 and Ufd3/Doa1. Given the huge error bars in these figures, the authors should perform statistical analyses to show whether the difference is significant.

*Reviewer #2:*

The authors have responded to my major concerns in an adequate manner, and the quality of the manuscript has been significantly improved.

*Reviewer #3:*

In their revised manuscript, Escobar-Henriques and co-workers have addressed all my concerns appropriately. The somewhat counter-intuitive observation that UBP12 deletion stabilises Ubp2p (and Fzo1p) is now sufficiently discussed. Concerns regarding over-expression of DUBs have been addressed. The Discussion is improved. The Materials and methods section now contains the requested tables. For these reasons, I support publication of the manuscript.

---

## [Author Response]

Essential revisions:While all three reviewers find the results potentially interesting the following points must be addressed to solidify the key conclusion of this study.1) The main claims of this manuscript rely on the degradation of DUBs under strong overexpression. Thus, the evaluation of Ubp2 and Ubp12 half-lives should be re-analyzed and characterized under endogenous expression.

We now analyzed the turnover of endogenous Ubp12 and Ubp2. We confirm their instability and show that their degradation depends on Cdc48-for Ubp12 (Figure 3) and on Ubp12-for Ubp2 (Figure 5).

2) The negative regulation of Ubp2 by Ubp12 is a central point of model proposed but not fully characterized. In principle the removal of ubiquitin from a substrate should increase and not decrease its stability so the authors should clarify how Ubp2 is destabilized by Ubp12.

Indeed, Ubp12 does not stabilise Ubp2, Fzo1, Rad23 (Gödderz JCS 2017) and Gpa1 (Wang JBC 2005), i.e. all substrates known so far. We tackled this particularity using Fzo1. We find that Ubp12 recognizes ubiquitylated forms on Fzo1 that only contain a very small number of ubiquitin moieties, which might explain why they do not serve as a good signal for proteasomal degradation (Figure 6). Consistently, the ubiquitin signals that accumulate in all Ubp12 substrates are composed of a limited number of discrete bands, instead of the high molecular weight smear, typical for polyubiquitylated substrates that are degraded by the proteasome. In the case of the protein Met4, also with short ubiquitin chains, these are shielded and therefore not recognized for degradation. It would be interesting to obtain a crystal structure of Ubp12 to analyse why it cleaves non-degradative ubiquitin signals.

3) The statistical analysis of numerous data should be improved.

We now show standard deviations in all quantifications and additionally provide statistically analysis when required.

New experimental data is provided in Figure 3, in Figure 5 and in the new Figure 6; Figure 1—figure supplement 1, Figure 3—figure supplement 1, Figure 4—figure supplement 2 and D and Figure 6—figure supplement 1. Moreover, new statistical analysis is presented in Figure 1, Figure 4, Figure 8 (old 7C) and in Figure 4—figure supplement 2 (old Figure 4—figure supplement 1). In addition, we present an extended Figure 2 (old 2C) and improved the quality of Figure 5 (old 5D). Finally, as requested, we have included tables describing the yeast strains, plasmids and antibodies used.

The comments of the individual reviewers are pasted below. You might want to address the additional concerns of the reviewers in case you can.Reviewer #1:[…] 1) The conclusions of this study rely heavily on overexpression of tagged deubiquitylases as the substrate of Cdc48. Considering the reported role of Cdc48 in diverse protein quality control processes, the authors need to exclude the possibility that overexpressed Ubp12 and Ubp2 are short-lived because of a potential misfolding issue caused by protein overexpression. One way to address this is to tag these DUBs endogenously and analyze their stability in wild type and Cdc48 mutant cells by cycloheximide chase.

Both Ubp12 and Ubp2 have now been genomically tagged. Cycloheximide chase experiments revealed that both Ubp12 and Ubp2 are unstable proteins (Figure 3 and Figure 5). Moreover, Ubp12 degradation depended on Cdc48 and Ubp2 degradation depended on Ubp12.

2) In Figure 2, the authors show an interaction between Cdc48 and Fzo1 that is dependent on Fzo1 ubiquitination. However, it is unclear what is the functional consequence of this interaction.

As for the functional consequence of Cdc48-Fzo1 interaction, in the Discussion chapter we propose that a local regulation of Fzo1 by Cdc48 allows to increase the efficiency of the Cdc48-DUB cascade on Fzo1 regulation.

Is it also unclear whether the interaction is direct or indirect.

In principle, there is no need for another mediator in the co-immunoprecipitation between Cdc48 and Fzo1, because Cdc48 binds ubiquitylated substrates and Fzo1 is ubiquitylated. However, if the physical interaction between Fzo1 and Cdc48 would be indirect, the best candidate to mediate it would be Ubp12. On the new Figure 4—figure supplement 2, we show that this is not the case. Nevertheless, we cannot exclude an indirect interaction between Cdc48 and Fzo1.

In the subsection “Cdc48 supports turnover of ubiquitylated Ubp12”, the conclusion that Cdc48 specifically interacts with pro-fusion ubiquitin is not supported by any data.

We show that Cdc48 binds to Fzo1 depending on its pro-fusion forms (Figure 2) and also show that the additional presence of the anti-fusion forms does not increase binding of Cdc48 (Figure 2). Moreover, we now show that the exclusive presence of the anti-fusion bands does not allow Cdc48 binding to Fzo1 (Figure 2 – extended from the old Figure 2). These data support the specificity of Cdc48 for the pro-fusion bands. We have also revised the corresponding text, to make this point clear (subsection “Cdc48 supports turnover of ubiquitylated Ubp12”).

3) In Figure 3, the authors try to demonstrate that Ubp12 is an unstable protein that is degraded in a Cdc48 dependent manner. All the experiments were done with overexpressed Ubp12. The authors should analyze the stability of endogenous Ubp12. In addition, several experiments shown are incomplete (e.g. Figure 3, also Figure 5). These experiments should include a cdc48 mutant for comparison.

We now show turnover of endogenous Ubp12 (Figure 3). We also now included a Cdc48 mutant for comparison (Figure 3 and Figure 3—figure supplement 1).

4) Figure 6 shows that the stability of Ubp2 is regulated by Ubp12. Once again, only overexpressed Ubp2 was studied. It is also puzzling why loss of Ubp12 can stabilize Ubp2, because DUB deficiency usually leads to increased ubiquitination and turnover. The quality of Figure 5 needs to be improved.

The stability of genomic Ubp2 in dependence of Ubp12 has now been analyzed (Figure 5) and the quality of Figure 5 had been improved by providing lower exposed blots. We agree that DUB deficiency usually leads to increased ubiquitylation and turnover. However, this is not the case for any of the four known substrates of Ubp12 – Fzo1, Ubp2, Rad23 (Gödderz JCS 2017) and Gpa1 (Wang JBC 2005). As discussed below, on point 5, we propose that those ubiquitylated forms, recognized by Ubp12, are too short to be a good proteasomal tag.

5) The authors argue that there are two forms of ubiquitinated Fzo1; one supports mitochondrial fusion while the other mitigates it, but the nature of the pro-fusion ubiquitination is unclear. The authors should consider analyzing the ubiquitin linkages on Fzo1 that is accumulated in Ubp12 mutant cells. Are Ubp2 and Fzo1 conjugated with the same type of ubiquitin chains preferred by Ubp12?

We show now, in Figure 6, that both Fzo1 and Ubp2 accumulate K48-linked chains in D*ubp12* cells.

This is an important issue because it is a bit hard to understand how Ubp12 can act on two different kinds of ubiquitin signals; the one on Fzo1 is incompetent for degradation, but the one on Ubp2 seems to influence its stability in a way that is counter-intuitive.

We apologize that the effect of Ubp12 on Fzo1 and on Ubp2 might have been unclear in the previous version. We now show that the ubiquitin signals that Ubp12 recognizes on Fzo1 resemble the ones that Ubp12 recognizes on Ubp2: in both cases it slows down their turnover (Figure 5 and Figure 6—figure supplement 1). We agree that this is counter-intuitive and therefore have further investigated it. In fact, the ubiquitin signals that accumulate in Fzo1, Ubp2, Rad23 and Gpa1 are all composed of a limited number of discrete bands, instead of the high molecular weight smear, typical for polyubiquitylated substrates. We have explored this for Fzo1 and found it to be composed of a di-ubiquitin chain (Figure 6). This might explain why accumulation of these chains, in Δ*ubp12* cells, does not increase substrate turnover.

Thus, in my opinion, to make the model convincing, the authors need to characterize the Ubp2-mediated deubiquitination reaction in more detail. What kind of linkage is preferred by Ubp12? Are Ubp2 and Fzo1 conjugated with the same type of ubiquitin chains preferred by Ubp12? Do these chains increase protein stability as oppose to target them for degradation?

We have performed experiments analyzing the chain preference of Ubp12. Ubp12 revealed to be very active and unspecific, presenting no preference for long *vs.* short ubiquitin chains, and also equally cutting K48 or K63-linked chains (Figure 6). So, we propose that the presence of short ubiquitin chains on its substrates explains why they are not targeted for the UPS.

6) "Fzo1 ubiquitination requires its lysine 464." If Fzo1 carries two types of ubiquitin chains, are they both attached to lysine 464? If yes, how are these processes coordinated? If not, where is the pro-fusion ubiquitin chain attached?

Fzo1 is ubiquitylated at lysines 464 and 398. Fzo1 is first ubiquitylated at lysine 464 and then induces the formation of pro-fusion ubiquitin chains on lysine 398. Therefore, upon mutation of lysine 464 all pro-fusion ubiquitylation is lost (Anton 2013 Mol Cell). We have now briefly introduced this in the second chapter of the Results, subsection “Cdc48 binds and regulates ubiquitylated Fzo1”.

Reviewer #2:[…] 1) Data in Figure 5 suggest that a DUB, Ubp12, is required for degradation of Ubp2. The authors show that Ubp2 is ubiquitylated and stabilized in the absence of Ubp12. In the presence of Ubp12, the ubiquitylated form of Ubp2 cannot be found and Ubp2 is degraded by the proteasome. In other words, the authors propose a model in which deubiquitylation of Ubp2 by Ubp12 promotes its degradation by the proteasome. It is very surprising that removal of ubiquitin from a substrate, rather than its addition to it, triggers proteasome-dependent degradation. In my eyes, this requires further explanations, ideally with additional experiments. I consider this an essential point, because the negative regulation of Ubp2 by Ubp12 is a central point of the authors' model.

We agree that DUB deficiency usually leads to increased ubiquitylation and turnover. In contrast, Ubp12 deficiency decreases turnover of Ubp2 and Fzo1 (Figure 5 and Figure 6—figure supplement 1). We have addressed this point by performing additional experiments on Fzo1. We found that the ubiquitin forms bound by Ubp12 are composed of a di- ubiquitin chain (Figure 6). Therefore, we propose that they are too short to be a good signal for proteasomal degradation. So far there are 4 substrates known for Ubp12: Fzo1, Ubp2, Rad23 (Gödderz JCS 2017) and Gpa1 (Wang JBC 2005). Importantly, in all these 4 substrates, the ubiquitin signals that accumulate in Δ*ubp12* cells are composed of a limited number of discrete bands, instead of the high molecular weight smear, typical for polyubiquitylated substrates. Consistently, Ubp12 also does not stabilize any of these 4 substrates.

Additional experiments analyzing Ubp12 have been performed. First, we show that Ubp12 regulates K48-linked chains on both Fzo1 and Ubp2 (Figure 6). Second, we analyzed the DUB activity of Ubp12 in vitro. Ubp12 revealed to be very active and unspecific, presenting no preference for long *vs.* short ubiquitin chains, and also equally cutting K48 or K63-linked chains (Figure 6). So, we propose that the presence of short ubiquitin chains on its substrates explains why they are not targeted for the UPS.

2) The term "ubiquitin homeostasis" should be removed from the title, as this has not been addressed in the paper.

Ubiquitin homeostasis has been deleted from the title.

3) The authors did not use any statistical analysis (like Student's t-test or ANOVA) to substantiate their numerous quantifications. Moreover, they used standard errors instead of standard deviations for error bars. Since standard errors underrepresent the variance of data the authors have to show all their quantitative data with standard deviation. Alternatively, they can keep standard errors as error bars but must then provide appropriate statistical analysis.

We now present standard deviations in all graphs.

In particular, the effects shown in Figure 4, Figure 4—figure supplement 1 and Figure 7 are rather moderate and must be substantiated by a more rigorous statistical analysis. If these results turn out to be not significant, the associated statements and conclusions have to be toned down accordingly.

Statistical analysis has now been provided for these three panels.

Reviewer #3:[…] 1) An important finding of this work is that the DUB Ubp12 is degraded in a Cdc48-dependent manner. Data for this is presented in Figure 3. In these experiments (and in others in which a DUB is expressed from a plasmid), Ubp12 is strongly overexpressed from a 2µ plasmid with the strong ADH promoter (as described in Anton et al., 2013). It is possible that degradation occurs because of the overexpression, e.g. because of then sub-stoichiometric amounts of a binding partner that normally stabilizes Ubp12. There are many examples in the literature for the degradation of orphan subunits in the literature, that are otherwise stable proteins (e.g. 1: Braun and Jentsch, 2007, EMBO Rep. 8(12):1176-82; 2: Habeck et al., 2015, JCB 209(2):261-73). I suggest that either this experiment is repeated using a chromosomally tagged version of Ubp12 under its endogenous promoter or that the possibility of an artifact due to overexpression is at least discussed. The latter would suffice in my opinion because the interplay of Cdc48 and Ubp12 is nicely shown in the following Figure 4 and its supplement.

This essential point was analyzed and is also now addressed in the Results section. Importantly, we could confirm that genomically expressed Ubp12 is an unstable protein, and that its turnover depends on Cdc48 (Figure 3).

2) In Figure 2, the authors provide evidence that Cdc48 is physically interacting with Fzo1, depending on ubiquitin(-chains) on K464. What is the proposed function of this interaction?

It is true that our Cdc48-Ubp12-Fzo1 regulon does not explain why Cdc48 would need to bind to Fzo1 in order to regulate it. However, we now propose in the Discussion section that a local regulation of Fzo1 by Cdc48 -*via* Ubp12- allows to increase the efficiency of the Cdc48-DUB cascade on Fzo1 regulation.

3) Cdc48 seems to have other functions in this pathway, apart from regulating Ubp12, since steady state levels of Fzo1 are not restored in cdc48-2/ ubc12Delta cells. Please comment.

This is absolutely correct and has now been further examined (Figure 4—figure supplement 2) and clearly stated in the respective Results section (subsection “Cdc48 regulation of Fzo1 depends on Ubp12”, corresponding to Figure 4—figure supplement E).

4) Figure 4—figure supplement 1 are insufficiently explained. Especially panel D is difficult to understand since Dnm1 is not introduced and the rationale for the experiment is missing.

We apologize for this confusion and have now provided an explanation in the Results subsection “Cdc48 regulation of Fzo1 depends on Ubp12”. Briefly, in absence of Fzo1 no hypertubulation has been observed upon deletion of *UBP12*. However, in Δ*fzo1* cells mitochondria are no longer tubular. We wanted therefore to exclude that the effect of Ubp12 depends on the shape of mitochondria, rather than on the role of Ubp12 in Fzo1. To this aim, the effect of Ubp12 was tested in Δ*fzo1* Δ*dnm1* cells, lacking Fzo1 but resembling wt cells in mitochondrial shape, so that the starting point would be tubular mitochondria. However, deletion of *UBP12* did not alter mitochondrial morphology of Δ*fzo1* Δ*dnm1* cells (Figure 4—figure supplement 1). This proves that in absence of *FZO1*, even tubular mitochondria cannot be altered by Ubp12. So, Ubp12 regulates mitochondrial morphology *via* Fzo1.

5) A table providing information on plasmids used in this study would be helpful, especially since information about expression of DUBs (overexpression, ADH promoter) is somewhat hidden by merely referencing Anton et al. (2013).

New tables describing the plasmids, strains and antibodies used are now provided.

6) The discussion about a role of Cdc48 in membrane fusion remains rather superficial. The proposed mechanism for the role of ubiquitination in Syntaxin 5-mediated fusion is rather different, namely that it prevents SNARE pairing. Here, p97 would mediate deubiquitination of Syn5 and thereby activate Syntaxin 5 (Huang et al. 2016). Furthermore, there is otherwise little evidence for a "general role of Cdc48" in membrane fusion. I suggest rephrasing of this paragraph.

We have proceeded as requested by the reviewer 3 (subsection “Roles of Cdc48 on mitochondrial dynamics”).

[Editors' note: further revisions were requested prior to acceptance, as described below.]

The manuscript has been improved but there are some remaining issues that need to be addressed before acceptance, as outlined below:The concerns from Reviewer 1 should be addressed in full. In particular, given the small changes in the turnover of endogenous Ubp2 (Figure 5) it is important to test whether they are statistically significant.

Figure 5 has been updated with the statistical analysis, showing that the changes in the turnover of endogenous Ubp2 are statistically significant.

Moreover, the authors should discuss alternative mechanisms of Ubp2 regulation and that may be independent of its proteolysis.

Possible alternative mechanisms of Ubp2 regulation, independent of its proteolysis, have now been discussed in the subsection “Regulation of DUB activity by ubiquitin”.

In brief, besides protecting from degradation, ubiquitylation of Ubp2 could also influence its localization and/or activity by several means, e.g.:

- allowing the formation of protein complexes which favor better activity, Cdc48 would be a candidate;

- inducing conformational changes favoring binding of Ubp2 to its substrates.

As requested, new statistical analysis is presented in Figure 5 and new experimental evidence and corresponding statistical analysis is presented in Figure 1—figure supplement 1.

Reviewer #1:This is a revised manuscript that addresses the role of Cdc48, an AAA ATPase in mitochondria dynamics using yeast as a model. In the first-round review, a major problem identified by all referees is that the analyses of protein stability are based entirely on overexpressed Usp12 and Usp2. The authors now tagged Ubp12 and Ubp2 endogenously and analyzed their turnover in different genetic backgrounds. However, the newly collected data do not seem to support the authors’ main conclusions.1) Subsection “Cdc48 supports turnover of ubiquitylated Ubp12” – the authors concluded that the stability of endogenous Ubp12 is regulated by Cdc48. However, the phenotype in my opinion is quite weak (Figure 3). Although the authors provide quantification results, the representative gel does not convince me that endogenous Ubp12 is unstable., The turnover is not as obvious as that of overexpressed Ubp12 (Figure 3—figure supplement 1), suggesting that overexpressed Ubp12 may not be properly folded and thus becomes a Cdc48 substrate.

We agree that the turnover rate is higher for the overexpressed Ubp12 than for the endogenous protein, as shown in the quantifications presented in Figure 3 and Figure 3—figure supplement 1. Nevertheless, upon Cdc48 impairment, the stabilization of endogenous Ubp12 is much more obvious than the one presented by overexpressed Ubp12. This convinces us that Ubp12 is a Cdc48 substrate, independently of its expression levels.

Compared to Ubc6, a previously documented Cdc48 substrate, the accumulation of Ubp12 in cdc48-2 mutant cells is marginal. Given that there may be only a small increase in Ubp12 protein level in cdc48-2 mutant cells, I am not convinced that the role of Cdc48 in regulation of mitochondria dynamics is achieved through controlling the stability of Ubp12.

The observation of a smaller increase in Ubp12 endogenous protein level in *cdc48-2* mutant cells is expected, given the lower degradation kinetics discussed above. Ubc6 was used here just as a positive control of the CHX chase experiments. We do not intend to compare different Cdc48 substrates, which of course can have very different turnover rates.

Moreover, although the turnover rate of Ubp12 is low, as previously pointed out by reviewer 1, we provide strong genetic data demonstrating that Cdc48 depends on Ubp12 for the regulation of mitochondrial morphology and cellular respiration.

2) Another major conclusion of the study is that the two DUBs form a regulatory "cascade" with the stability of Ubp2 being controlled by Ubp12. The new result shown in Figure 5 does show that endogenous Ubp2 is unstable, but intriguingly, although the degradation of Ubp2 appears to be inhibited when Ubp12 gene was deleted, there is no obvious accumulation of Ubp2 in ubp12 deficient cells. Thus, it is unclear how Ubp12 could regulate the stability of Fzo1 in a Ubp2 dependent manner.

We now present statistical analysis clearly demonstrating that Ubp2 is an unstable protein, and that its turnover is reduced in Δ*ubp12* cells. We would like to point out that the differences in loading of the Ubp2 CHX chase in wt and Δ*ubp12* cells, in Figure 5, visible from both the Ssc1 and Ubc6 signal intensities, indeed give the impression that Ubp2 does not accumulate in the absence of *UBP12*.

It is correct that Ubp12 only partially affects Ubp2 turnover rates. So, as suggested, we propose now in the Discussion subsection “Regulation of DUB activity by ubiquitin” that ubiquitylation could additionally regulate Ubp2 by proteolysis-independent means. For example, it could increase its activity by favoring further PTMs, consistent with the identification of phosphorylated sites on Ubp2 (Swaney, 2013). Moreover, Ubp2 ubiquitylation could favor allosteric conformational changes. This could occur autocatalytically – supported by the observation that Ubp2 is among the largest yeast DUBs – or as part of a protein complex with Cdc48, in agreement with our observation that Cdc48 and Ubp2 co-immunoprecipitate.

Other issues:3) Subsection “Cdc48 promotes mitochondrial fusion and prevents Fzo1 turnover” – "Consistent with…". The authors conclude that the levels of Fzo1 were slightly decreased in the cdc48-3 mutant or in cells deleted for the Cdc48 co-factor factors Npl4, Ufd1 and Ufd3/Doa1. Given the huge error bars in these figures, the authors should perform statistical analyses to show whether the difference is significant.

After performing more replicates, we can show now that the differences of Fzo1 levels in the mutant cells for Cdc48 and co-factors Npl4, Ufd1 and Ufd3/Doa1, are statistical significant (Figure 1—figure supplement 1). This supports our previous conclusions.